# Structural characterisation of the fungal Pmt4 homodimer

Melanie A. McDowell [1,6] ✉, Klemens Wild [1], Francesco Fiorentino [2,3], Daniela Bausewein[4], Anke Metschies[4], Antonella Chiapparino[1], Yvonne Hackmann[1], Florestan L. Bilsing[1], David Brenske[4], Sofia Mortensen[1], Di Wu [3], Carol V. Robinson [3,5], Sabine Strahl[4] & Irmgard Sinning [1] ✉

Protein O-mannosyltransferases (PMTs) are conserved endoplasmic reticulum membrane-embedded enzymes responsible for the transfer of mannose from dolichol phosphate-mannose (Dol-P-Man) to serine/threonine-rich protein substrates or unfolded proteins. PMTs from three subfamilies form obligate dimers with different substrate specificities and require the concerted action of their transmembrane domains (TMDs) and a luminal MIR domain for catalysis. Here, we present structures, native mass spectrometry, and structure-based mutagenesis of the fungal Pmt4 homodimer. The core fold of the TMDs and MIR domain is conserved with the Pmt1-Pmt2 heterodimer, indicating a shared catalytic mechanism. Distinct from Pmt4, the MIR domain interacts in cis with the TMDs of the same subunit and has a β-hairpin insertion required for O-mannosylation of substrates. We further identify a cytosolic binding site for substrate Dol-P-Man within the Pmt4 TMDs, which is conserved amongst PMTs and important for in vivo activity. Thus, we provide a framework to understand the substrate specificity and regulation of the Pmt4 homodimer.

Protein O-mannosylation is an evolutionarily conserved and essential post-translational modification that occurs in the endoplasmic reticulum (ER) of eukaryotes[1]. In yeast, the majority of proteins entering the secretory pathway (including 90% of all cell wall proteins) are O-mannosylated[2] and unfolded protein O-mannosylation (UPOM) is important for ER protein quality control[3], whilst impairment of O-mannosylation in humans leads to congenital muscular dystrophies[4]. The transfer of mannose from dolichol monophosphate-activated mannose (Dol-P-Man) to the hydroxyl group of serine/threonine residues of substrate proteins is catalysed by membrane-embedded glycosyltransferases of the GT-C superfamily[5]. The most widely conserved family is the protein O-mannosyltransferases (PMT in yeast, POMT in mammals), which target largely unstructured regions within a range of substrates[1,2]. More recently, mammalian TMCT1-4 and TMEM260 enzymes were also found to O-mannosylate specific folded domains[6].

In *Saccharomyces cerevisiae*, six active PMT proteins are classified into three families on the basis of sequence similarity (Supplementary Fig. 1): PMT1 (Pmt1 and Pmt5), PMT2 (Pmt2, Pmt3 and Pmt6) and PMT4 (Pmt4)[7]. These proteins form obligate dimers within or across families[8] and, despite a degree of functional redundancy, a Pmt1-Pmt2 heterodimer and Pmt4-Pmt4 homodimer account for the majority of O-mannosylation[9]. In contrast, a heterodimer comprising POMT1 and POMT2 (PMT4 and PMT2 family members, respectively) is the sole PMT-like O-mannosyltransferase present in mammals[10].

The cryo-EM structure of the *S. cerevisiae* (sc) Pmt1-Pmt2 heterodimer, involved in both UPOM and O-mannosylation of bona fide S/T-rich substrates of the secretory pathway, revealed each subunit to comprise 11 transmembrane domains (TMDs) that share a fold with the catalytic subunits of oligosaccharyltransferases[11] and tryptophan C-mannosyltransferases[12]. A further structure of the *sc*Pmt1-Pmt2

[1]Heidelberg University Biochemistry Centre (BZH), Heidelberg, Germany. [2]Department of Biochemical Sciences, Sapienza University Rome, Rome, Italy. [3]Department of Chemistry, University of Oxford, Oxford, UK. [4]Centre for Organismal Studies (COS), Heidelberg, Germany. [5]Kavli Institute for NanoScience Discovery, Oxford, UK. [6]Present address: Membrane Protein Biogenesis Research Group, Max Planck Institute of Biophysics, Frankfurt am Main, Germany. ✉e-mail: melanie.mcdowell@biophys.mpg.de; irmi.sinning@bzh.uni-heidelberg.de

complex bound to the donor Dol-P and acceptor peptide confirmed that the active site is located in a luminal pocket of each subunit, where an invariant DE motif within the loop between TMD1 and TMD2 catalyses transfer of the mannose by an inverting mechanism. Aside from the transmembrane region, PMT proteins contain a luminal MIR domain between TMD7 and TMD8, so-called due to their presence in mannosyltransferases, inositol triphosphate receptors and ryanodine receptors[13]. Within the scPmt1-Pmt2 structure, the scPmt1-MIR domain interacts in trans with the TMDs of scPmt2 next to the active site and the scPmt2-MIR domain is disengaged[11], although the dynamic nature of these interactions cannot be inferred. Furthermore, the characteristic β-trefoil fold of PMT2-MIR domains was shown to comprise mannose-specific binding sites, which are in part conserved with carbohydrate-binding modules (CBMs) and important for O-mannosylation of S/T-rich peptides[14]. This led to a model whereby the MIR domain binds to a mannosylated peptide product to keep it away from the adjoining PMT active site and ensure enzyme processivity.

Currently, there is no structural information available for the PMT4 family of O-mannosyltransferases and it cannot be inferred from the scPmt1-Pmt2 structure how the MIR domains would interact with the TMDs within the Pmt4-Pmt4 homodimer. In contrast to the recognition of both soluble and membrane proteins by Pmt1-Pmt2 and its additional involvement in UPOM[15], Pmt4 predominantly mannosylates S/T-rich regions in membrane protein substrates[9,16,17], however, the molecular basis of this substrate specificity is also unknown. In addition, yeast Pmt4 provides a powerful model for understanding the significant group of congenital muscular dystrophies linked to reduced O-mannosylation of α-dystroglycan (αDG) by POMT1-POMT2, as it mannosylates αDG-derived peptides and is catalytically impaired by the insertion of pathogenic POMT1 mutations[18].

Here, we use a combination of structural biology, native mass spectrometry and in vivo functional assays to characterise the fungal Pmt4 homodimer. The MIR domain of Pmt4 interacts in cis with the TMDs of the same subunit and contains a unique β-hairpin insertion important for O-mannosylation of substrates. The structure of Dol-P bound within the Pmt4 luminal active site highlights a strictly conserved catalytic mechanism between PMT families. Surprisingly, our data reveal an additional binding site for Dol-P-Man on the cytosolic side of the membrane, which we show to be important for the O-mannosyltransferase activity of Pmt4.

## Results

### The structure of the Pmt4 homodimer
To investigate the structure of Pmt4, we took advantage of its thermostable homologue from the thermophilic fungus *Chaetomium thermophilum*, which has 48 % sequence identity to yeast Pmt4 (Supplementary Fig. 1). *C. thermophilum* Pmt4 (*ct*Pmt4) was overexpressed in *S. cerevisiae* and purified via a C-terminal deca-histidine tag in detergent lauryl maltose neopentyl glycol (LMNG) supplemented with cholesteryl hemisuccinate (CHS) (Supplementary Fig. 2A). Structural analysis by single-particle cryo-electron microscopy (cryo-EM) confirmed *ct*Pmt4 is a homodimer (Fig. 1A), allowing processing of the data with either no symmetry (C1) or two-fold rotational symmetry (C2) imposed (Supplementary Fig. 3A, B and Table 1). Within the membrane region, the 3.2 Å C2 map showed the same features as the 3.4 Å C1 map but at higher resolution (Supplementary Fig. 3C), and was therefore used to build a two-fold symmetric model for the *ct*Pmt4 transmembrane domains (TMDs) (Supplementary Fig. 4A). In contrast, the luminal MIR domains were arranged asymmetrically and only resolved in the C1 reconstruction at a resolution that precluded direct model building (Supplementary Fig. 3C). Therefore, we purified the *ct*Pmt4-MIR domain from *Escherichia coli* via a N-terminal hexa-histidine tag (Supplementary Fig. 2B) and solved the crystal structure at 1.2 Å resolution by molecular replacement (Table 2) (Supplementary

Fig. 4B). Our final composite model for the *ct*Pmt4 homodimer (Fig. 1A, B) was derived from rigid body fitting these high-resolution structures for the TMDs and MIR domain into the C1 reconstruction and directly building models for their connecting linkers into the remaining density.

The 11 TMDs of *ct*Pmt4 have a highly similar arrangement to those of *sc*Pmt1 and *sc*Pmt2[11], superimposing with a root-mean-square deviation (RMSD) of 1.6 Å and 1.4 Å respectively (Fig. 1C). Moreover, the *ct*Pmt4 homodimer adopts the same overall morphology within the membrane region as the *sc*Pmt1-Pmt2 heterodimer, with a rhombic cavity separating the dimer halves (Fig. 1B). As in *sc*Pmt1-Pmt2, residues at the cytosolic ends of TMD6 (W263 and L266) and TMD8 (R629 and R630) form the major dimer interface, bounding the rhombic cavity on two sides at the cytosolic side of the membrane (Fig. 1D). However, the luminal ends of TMD9 from each dimer half are tilted further towards each other, resulting in an additional single constriction point at F662 and V663 and a smaller rhombic cavity than for *sc*Pmt1-Pmt2 (Fig. 1B, D). Interestingly, the residues at these dimer interfaces are largely conserved (Supplementary Fig. 1) and thus do not provide a satisfactory explanation for highly specific Pmt4-Pmt4 or Pmt1-Pmt2 dimer formation. Therefore, the N-terminus of *sc*Pmt1 and the interfacial lipid, which additionally contribute to the *sc*Pmt1-Pmt2 dimer interface[11] and are not resolved in the *ct*Pmt4 map, could be key elements that drive exclusive heterodimer formation.

The luminal MIR domains of *ct*Pmt4 adopt a strikingly different arrangement with respect to the TMDs, compared to within the *sc*Pmt1-Pmt2 heterodimer; whilst the *sc*Pmt1- and *sc*Pmt2-MIR domains are positioned to interact in trans with the TMDs of the opposing dimer subunit, the first *ct*Pmt4-MIR domain is rotated by 30° relative to the membrane and consequently interacts in cis with the TMDs from the same dimer subunit (Fig. 1B, C). Interestingly, whilst the second *ct*Pmt4-MIR domain is also positioned in cis, it is rotated a further 30° so that it no longer interacts with the TMDs, akin to the *sc*Pmt2-MIR domain. Therefore, despite major differences in the cis/trans nature of their interactions, the MIR domains of both the *ct*Pmt4 homodimer and *sc*Pmt1-Pmt2 heterodimer have a similar asymmetric arrangement, whereby one MIR domain is attached to the TMDs and the other is detached from them (Fig. 1B). As a result of this cis arrangement of the *ct*Pmt4-MIR domains, the homodimer is maintained solely by the interactions defining the rhombic membrane cavity and perhaps explains why an additional luminal contact is present (Fig. 1D).

### Pmt4-MIR domains have a functionally important β-hairpin insertion
Previously, high-resolution crystal structures were solved and functionally annotated for MIR domains from *S. cerevisiae* PMT2 family members[14]. Therefore, in order to be able to compare PMT2 and PMT4 family MIR domains from the same species, we additionally purified the *sc*Pmt4-MIR domain (Supplementary Fig. 2C) and solved the crystal structure to 1.3 Å resolution (Table 2) (Supplementary Fig. 4B). Aside from minor conformational differences in the β4−β5 and β8−β9 loops, the *sc*Pmt4- and *ct*Pmt4-MIR domain structures are highly similar (RMSD 0.87 Å over 194 Cα-atoms) (Fig. 2A) and comprise the characteristic peculiar β-trefoil fold i.e. six β-hairpins arranged within a pseudo-threefold symmetry (three MIR-motifs: MIRm1,2,3), altogether forming a β-barrel and a triangular cap (Fig. 2B).

Overall, the *sc*Pmt4-MIR domain closely resembles *sc*Pmt2 in its core fold (RMSD 1.38 Å over 168 Cα-atoms) (Fig. 2C) and accordingly putative binding sites for O-mannosylated peptides could also be located in each MIR-motif (α, β, γ and δ) (Fig. 2B). The highly conserved LH(S/T)H and DxNN (x: any residue) fingerprints in MIRm1 and MIRm2 define the α and β sites respectively (Supplementary Fig. 5), which are likely to form mannose-specific binding sites similar to other CBMs[19]. However, the fungal Pmt4 structures have a short three-residue

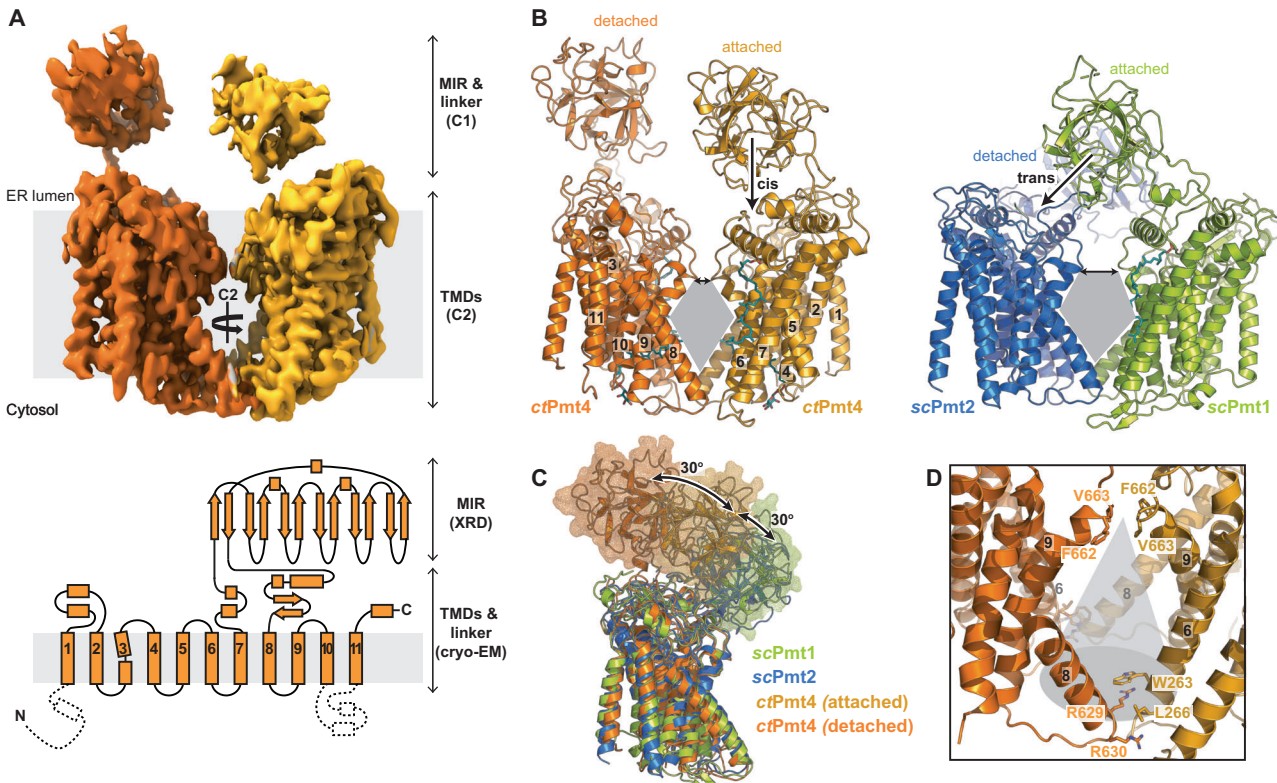

**Fig. 1 | The structure of the *ct*Pmt4 homodimer. A** The C1 cryo-EM reconstruction of the *ct*Pmt4 homodimer locally sharpened using LocScale[48] and coloured according to each dimer half. The membrane plane is defined perpendicular to the C2 symmetry axis of the TMDs, with boundaries inferred from the location of the TMDs. TMDs are numbered in the topology schematic (bottom), where dashed lines represent features not present in our model. The approximate positions of the MIR domains, linkers and TMDs are shown on both representations, together with the method used to determine their high-resolution structure (cryo-EM or X-ray diffraction (XRD)) and their rotational symmetry within the homodimer. **B** Side-by-side comparison of our model for the *ct*Pmt4 homodimer (left) with the model for the *sc*Pmt1·Pmt2 heterodimer[11] (PDB 6P25) after superimposition of the TMDs (RMSD 4.89 Å over 861 Cα-atoms). The TMDs of *ct*Pmt4 are numbered and the nature of the association of the MIR domains with the TMDs shown. The rhombic cavities are shown in grey and the luminal constrictions by double-headed arrows. **C** Superimposition of the TMDs of PMT homologues. The TMDs from each dimer half of *ct*Pmt4 were aligned, then superimposed with the TMDs of *sc*Pmt1 (RMSD 1.49 Å over 388 Cα-atoms) and *sc*Pmt2 (RMSD 1.33 Å over 330 Cα-atoms). Each subunit is coloured as in **B** and the surfaces of the MIR domains are shown in wireframe representation. The detached *sc*Pmt2-MIR domain is not shown. **D** The homodimer interface of *ct*Pmt4. The rhombic cavity is represented by a grey cone and the conserved residues mediating intersubunit contacts are labelled. For clarity, only one of the two symmetric contacts between the cytosolic ends of TMD6 and TMD8 are labelled.

insertion (*ct*: E449-Y451, *sc*: E442-D444) in MIRm2 relative to *sc*Pmt2 that is part of a unique α-helix (αMIR) of unknown function (Fig. 2B and Supplementary Fig. 5). Unlike MIRm1 and MIRm2, MIRm3 displays higher sequence divergence across PMT families and deviates from *sc*Pmt2 in exposed loop regions (Fig. 2C and Supplementary Fig. 5). Interestingly, whilst in the *sc*Pmt2-MIR the sugar-binding γ site of MIRm3 is sterically blocked, the γ site within the *sc*Pmt4-MIR is open and comprises a LWTH motif more similar to the α and β sites (Supplementary Fig. 5). In addition, the PMT4 family MIR domains (including the homologous human POMT1-MIR) do not contain the PMT2-MIRm3 δ site insertion[14]. Instead, three exposed loops, β1−β2, β8−β9 and β11−β12, are positioned around a central cavity that corresponds to the PMT2-MIRm3 δ site, which we altogether define as a distinct Pmt4-MIR δ site (Fig. 2B and Supplementary Fig. 5).

Characteristically, the Pmt4-MIR domain structures reveal a unique β-hairpin insertion (β-ins) of seven residues (RY(D/E)(D/N)GRΦ) in β2−β3 of MIRm1 (Fig. 2B, C and Supplementary Fig. 5). β-ins is extensively stabilised by internal hydrogen-bonding and a central salt-bridge between the conserved 4th and 6th residue (*ct*: D370:R372, *sc*: D364:R366) (Fig. 2D). It is fixed on top of the cap scaffold by hydrogen-bonding of the tyrosine (*ct*: Y368, *sc*: Y362) to a glutamate (*ct*: E441, *sc*: E434), and of the centrally-fixed arginine to the mainchain around a proline (*ct*: P434, *sc*: P427) within MIRm2. The tyrosine also stacks onto

this proline and is part of an aromatic cluster within the cap (including *ct*: Y364 and W501, *sc*: Y358 and W496).

To assess the importance of β-ins on the O-mannosyltransferase activity of Pmt4 in vivo, yeast strains were created that express either FLAG-tagged wild-type or mutated *sc*Pmt4 (*sc*Pmt4-FLAG), specifically with a deletion of β-ins (Δ362-368) or alanine substitutions of residues crucial for stabilising its fold (Y362A/D364A/R366 A (YxDxR), P427A/P430A (PP) and W496A) (Fig. 2D). Notably, using nano differential scanning fluorimetry (nanoDSF), we found that the isolated *sc*Pmt4-MIR domain with the most severe Δ362-368 mutation was folded, albeit with a slightly reduced stability relative to the wild-type domain (Supplementary Fig. 6A). All full-length *sc*Pmt4-FLAG variants were subsequently shown to be integrated in the membrane at the same level as wild-type *sc*Pmt4-FLAG (Supplementary Fig. 6B), allowing them to be tested using two assay systems. Firstly, we took advantage of the synthetic temperature sensitivity of the double deletion strain Δ*pmt1pmt4*, which only grows at 37 °C when complemented with enzymatically active *sc*Pmt4 (or *sc*Pmt1)[20]. Whilst wild-type and PP *sc*Pmt4-FLAG complement the temperature-sensitive phenotype, none of the other mutants restore thermal tolerance to Δ*pmt1pmt4* (Supplementary Fig. 6C), indicating that these *sc*Pmt4 variants have little or no mannosyltransferase activity. Secondly, we used the cell wall protein Ccw5 as a reporter, as it shows a markedly altered glycosylation

**Table 1 | Cryo-EM data collection, refinement and validation statistics**

|  | ctPmt4 C2 map (EMDB-52631) (PDB 9I5K) | ctPmt4 C1 map (EMDB-52632) (PDB 9I5L) |
|---|---|---|
| **Data collection and processing** | | |
| Detector | K2 | |
| Magnification | 165,000 | |
| Voltage (kV) | 300 | |
| Electron exposure (e–/Å$^2$) | 42.7 | |
| Defocus range (μm) | 0.8–2.0 | |
| Pixel size (Å) | 0.827 | |
| Symmetry imposed | C2 | C1 |
| Initial particle images (no.) | 2,199,173 | 2,199,173 |
| Final particle images (no.) | 210,070 | 347,202 |
| Map resolution (Å) | 3.2 | 3.4 |
| FSC threshold | 0.143 | 0.143 |
| Map resolution range (Å) | 2.9–5.9 | 3.0–7.4 |
| **Refinement** | | |
| Initial model used (PDB code) | 6P25 | 9I5K and 9FD1 (this study) |
| Model resolution (Å) | 2.9 | 3.7 |
| FSC threshold | 0.5 | 0.5 |
| Map sharpening $B$ factor (Å$^2$) | –117 | -123 |
| Model composition | | |
| Non-hydrogen atoms | 8276 | 11736 |
| Protein residues | 986 | 1402 |
| Ligands | DOP:2, DOL:2 | DOP:2, DOL:2 |
| $B$ factors (Å$^2$) | | |
| Protein | 44 | 35 |
| Ligands | 38 | 38 |
| R.m.s. deviations | | |
| Bond lengths (Å) | 0.006 | 0.007 |
| Bond angles (°) | 0.724 | 0.928 |
| Validation | | |
| MolProbity score | 1.84 | 1.81 |
| Clashscore | 8.59 | 9.39 |
| Poor rotamers (%) | 0.00 | 0.16 |
| Ramachandran plot | | |
| Favoured (%) | 94.56 | 95.55 |
| Allowed (%) | 5.44 | 4.45 |
| Disallowed (%) | 0.00 | 0.00 |

**Table 2 | Crystallographic data collection and refinement statistics (molecular replacement)**

|  | ctPmt4-MIR domain (PDB 9FD1) | scPmt4-MIR domain (PDB 9FD0) |
|---|---|---|
| **Data collection** | | |
| Wavelength (Å) | 0.9677 | 0.966 |
| Space group | I222 | P2$_1$2$_1$2$_1$ |
| Cell dimensions | | |
| a, b, c (Å) | 61.16, 100.63, 103.27 | 48.59, 56.77, 78.22 |
| α, β, γ (°) | 90.00, 90.00, 90.00 | 90.00, 90.00, 90.00 |
| Resolution (Å) | 31.90-1.22 (1.26–1.22)*# | 41.28-1.35 (1.40–1.35)* |
| Total reflections | 129,741 (960) | 93,109 (9186) |
| Unique reflections | 64,952 (497) | 47,974 (4721) |
| $R_{pim}$ | 0.026 (0.506) | 0.038 (0.336) |
| Mean $I / \sigma I$ | 22.83 (1.22) | 10.78 (2.32) |
| CC$_{1/2}$ | 0.995 (0.685) | 0.999 (0.529) |
| Completeness (%) | 68.62 (5.29) | 99.43 (99.56) |
| Multiplicity | 2.0 (1.9) | 1.9 (1.9) |
| Wilson B-factor (Å$^2$) | 12.1 | 11.1 |
| **Refinement** | | |
| Reflections used in refinement | 64,935 (496) | 47,967 (4721) |
| Reflections used for R$_{free}$ | 2003 (15) | 1745 (172) |
| $R_{work}$ | 0.112 (0.276) | 0.152 (0.324) |
| $R_{free}$ | 0.140 (0.270) | 0.190 (0.368) |
| CC$_{work}$ | 0.980 (0.827) | 0.973 (0.764) |
| CC$_{free}$ | 0.975 (0.698) | 0.953 (0.765) |
| Number of non-hydrogen atoms | 2160 | 2042 |
| Protein | 1714 | 1777 |
| Ligand/ion | 163 | 39 |
| Water | 373 | 226 |
| Protein residues | 204 | 218 |
| Average $B$-factor (Å$^2$) | 19.9 | 17.6 |
| Protein | 14.5 | 16.0 |
| Ligand/ion | 50.4 | 26.6 |
| Water | 38.4 | 28.6 |
| R.m.s. deviations | | |
| Bond lengths (Å) | 0.007 | 0.021 |
| Bond angles (°) | 1.14 | 1.84 |
| Clash score | 7.38 | 7.01 |
| Ramachandran plot | | |
| Favoured (%) | 98.51 | 98.15 |
| Allowed (%) | 1.49 | 1.85 |
| Disallowed (%) | 0.00 | 0.00 |
| Rotamer outliers (%) | 1.06 | 2.01 |

*Values in parentheses are for the highest-resolution shell.
#Anisotropic scaling.

pattern when scPmt4 activity is impaired[16]. In Δpmt4-CCW5 cells expressing wild-type scPmt4, O-mannosylation in a S/T-rich region prevents N-glycosylation of residue N114 and Ccw5 has an apparent molecular mass of ~40 kDa. In contrast, lack of scPmt4-based O-mannosylation allows N-glycosylation of Ccw5, resulting in a shift in apparent molecular mass to ~90–120 kDa. Consistent with the temperature-sensitive growth phenotype, the glycosylation pattern of Ccw5 isolated from yeast strains expressing scPmt4 β-ins mutant proteins Δ362-368, YxDxR and W496A is very similar to that of the null mutant pmt4 (Fig. 2E), demonstrating that the enzyme function is largely lost in vivo. Therefore, β-ins is generally required for O-mannosylation of substrate proteins by Pmt4.

As observed previously in the PMT2-MIR crystal structures[14], the Pmt4-MIR domains are extensively bound to small polar ligands present in the crystallisation solutions (acetate or HEPES) or cryo-protectant buffers (glycerol or ethylene glycol) (Fig. 2B and Supplementary Fig. 7A). In particular, the α and β sites, which were demonstrated for scPmt2 to bind O-mannosylated peptides[14], are again occupied by ligands in ctPmt4-MIR and scPmt4-MIR respectively, in-line with the expectation that they are also sugar binding sites. Given that both the open γ site and β-ins of ctPmt4-MIR are also bound to

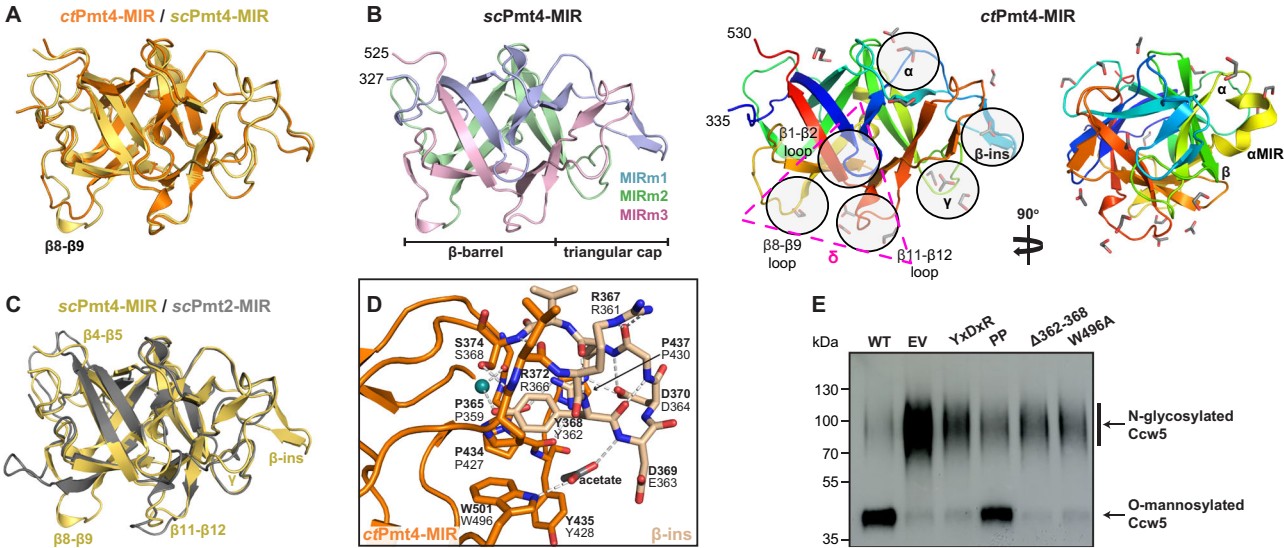

**Fig. 2 | Crystal structures of Pmt4-MIR domains. A** Superimposition of the *ct*Pmt4 and *sc*Pmt4-MIR domain crystal structures (RMSD 0.87 Å over 194 Cα-atoms) with the variable β8-β9 loop region shown. **B** Overall structures of the Pmt4-MIR domains from *S. cerevisiae* (left) and *C. thermophilum* (right) shown in the same view as **A**. *sc*Pmt4-MIR is coloured according to the MIR-motifs. *ct*Pmt4-MIR is coloured from the N-terminus (blue) to the C-terminus (red) and ligand binding sites (α to δ) are assigned as previously[14]. Unique structural features of Pmt4-MIR domains (αMIR and β-ins), along with the extended δ-site (magenta triangle), are also indicated. Bound acetate and ethylene glycol ligands present in the crystallisation solution are shown as sticks. **C** Superimposition of the *sc*Pmt4- and *sc*Pmt2-(PDB 6ZQP) MIR domains (RMSD 1.38 Å over 168 Cα-atoms) with variable loop regions indicated. **D** Key interactions stabilising β-ins (beige) within the *ct*Pmt4-MIR domain. *ct*Pmt4 residues are labelled in bold together with the corresponding residue in *sc*Pmt4. Hydrogen bonds are indicated by dashed lines, and an interacting acetate and water molecule (green sphere) are shown. **E** Western blot of Ccw5 isolated from a *S. cerevisiae* Δ*pmt4* strain expressing FLAG-tagged Ccw5 (Δ*pmt4-CCW5*) in combination with wild-type *sc*Pmt4 (pJK4-B1; WT), empty pRS423 vector (EV) and β-ins mutants of pJK4-B1 as indicated. Anti-FLAG antibody was used to detect O-mannosylated and N-glycosylated isoforms of Ccw5, which display distinct electrophoretic mobilities in a 10% SDS-PA gel. A representative blot is shown from three independent experiments. Source data are provided in the Source Data file.

acetate ligands (Fig. 2B, D and Supplementary Fig. 7A), these too may be mannose binding sites, which would indeed provide an explanation for the importance of β-ins in O-mannosylation by Pmt4. Furthermore, the *ct*Pmt4- and *sc*Pmt4-MIR domains have an asymmetric surface charge distribution with both a highly positively and negatively charged patch (Supplementary Fig. 7A). Strikingly, the bound ligands are aligned in a linear trace starting from the positively charged surface at the δ and γ sites, extending through β-ins and up to the α and β sites and the rather negatively charged surface (Supplementary Fig. 7A). This suggests a defined substrate/product peptide route, in-line with the model that PMT-MIR domains coordinate a mannosylated peptide to ensure a processive enzyme reaction[14].

The active site of PMTs is located in a deep cavity on the ER luminal side of the TMDs, and it has been postulated that the TMD-MIR interface plays a role in guiding substrates to or products away from the active site[11,14]. As in the *sc*Pmt1-Pmt2 heterodimer, the δ site of the *ct*Pmt4-MIR domain is positioned at the TMD-MIR interface, opposing the active site (Supplementary Fig. 7B). In the attached *ct*Pmt4 TMD-MIR interface, the active site is completely covered by this δ site, whilst in the detached conformation, the MIR domain is rolled aside and the active site is fully accessible (Supplementary Fig. 7C). However, PMT2-specific residue insertions in the δ and δ' sites, which in the *sc*Pmt1-Pmt2 heterodimer are primarily responsible for TMD-MIR contacts, are absent in PMT4 sequences[11,14]. Instead, the attached MIR domain directly contacts the TMDs via the β8−β9 loop, whereby exposed histidine and charged residues (E465, H466, K468) form a relatively weak interface with aromatic residues of the TMDs (F319, H561, W562) (Supplementary Fig. 7D). The rolling movement between the attached and detached state of the MIR domain is likely to be facilitated both by having this weak δ site interaction and the flexible ~20 residue linkers connecting the MIR domains and TMDs, which are arranged asymmetrically in our structure (Fig. 1A). Indeed, the poor resolution of the

*ct*Pmt4-MIR domains in the C1 cryo-EM reconstruction demonstrates that they are rather dynamically associated with the TMDs (Supplementary Fig. 3C). Although in the *sc*Pmt1-Pmt2 structure the MIR domains act in trans, their general orientation and surface properties are corresponding[11]. Thus, whilst the full trajectories and conformational space of the Pmt4-MIR domains are yet to be clarified, the alternate association with their TMDs likely acts as a general mechanism to regulate access or catalysis at the active site.

## The active site is highly conserved across PMT families

Our cryo-EM reconstructions revealed two single-chain lipid species bound within each membrane domain of the *ct*Pmt4 homodimer, arranged so that the hydrocarbon tails protrude into the central rhombic cavity (Fig. 3A and Supplementary Fig. 8A). The head group of one lipid species is positioned in the luminal membrane leaflet in the active site and was thus modelled as the reaction product Dol-P, as in the *sc*Pmt1 structure[11]. Similarly, only the first seven isoprene repeats of the dolichol chain are resolved in the density and these are accommodated within the membrane in a hydrophobic groove formed by residues from TMD5 (I241), TMD6 (Y245, L248, F251) and TMD7 (F284, I292, F296, Y299) (Supplementary Fig. 8B). The Dol-P phosphate group is also stabilised by the conserved residues H105, K244, R666 and H672 (Supplementary Figs. 1 and 8C), however is retracted 4.7 Å from the acceptor threonine binding site compared to within *sc*Pmt1 (Fig. 3B). Accordingly, the R666 side chain is rotated 60° to stabilise this alternative position of the Dol-P phosphate, similar to its position in *sc*Pmt2 where Dol-P was not bound (Supplementary Fig. 8C). Therefore, the dolichol product of the O-mannosylation reaction appears to be positioned closer to its putative release pathway. This implicit flexibility likely also explains why the density for the Dol-P phosphate group is consistently weaker than for the dolichol tail (Supplementary Fig. 8A)[11].

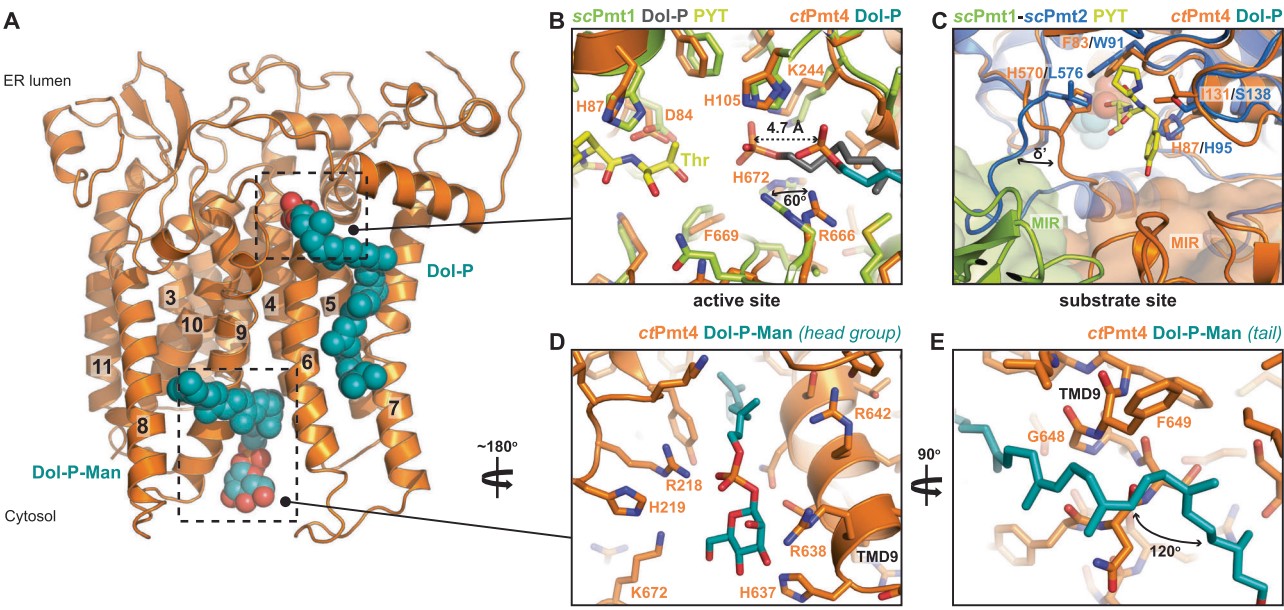

**Fig. 3 | Binding sites for dolichol lipids within *ct*Pmt4. A** Model for the *ct*Pmt4 TMDs viewed from the homodimer interface. Bound Dol-P and Dol-P-Man are shown in CPK representation and the TMDs are numbered. The dashed boxes correlate approximately to the detailed views shown in the other panels. **B** Detailed view of the active sites of *ct*Pmt4 and *sc*Pmt1 after superimposition as in Fig. 1C. The model substrate (PYT tri-peptide) bound to *sc*Pmt2 is also superimposed and the acceptor threonine residue labelled. Conserved residues involved in substrate/product binding are labelled for *ct*Pmt4. The distance between the bound Dol-P molecules in *ct*Pmt4 and *sc*Pmt1 is shown by a dashed arrow and the corresponding rotation of the conserved arginine side-chain by a solid arrow. **C** Detailed view of the *sc*Pmt2 PYT peptide binding site superimposed with *ct*Pmt4 as in Fig. 1C. Corresponding residues proximal to the peptide are labelled and a solid arrow shows the different conformations of the δ′ site. The attached MIR domains of *sc*Pmt1 and *ct*Pmt4 are shown in both cartoon and surface representation. Detailed view of the *ct*Pmt4 cytosolic binding site for the **D** Dol-P-Man headgroup and **E** dolichol tail. Residues involved in interactions with the phosphate or mannose groups or guiding the trajectory of the dolichol tail are labelled.

Superimposition of *ct*Pmt4 with both *sc*Pmt1 and *sc*Pmt2 indicates that the remaining residues for binding the donor dolichol and those expected to bind the acceptor threonine (F83, D84, H87 and F669) are conserved in their positions in the active site (Fig. 3B and Supplementary Fig. 8C). In addition, comparison with the bacterial oligosaccharyltransferase PglB[21] indicates that D84 and D197 are positioned appropriately for the coordination of a $Mn^{2+}$ ion (Supplementary Fig. 8D), however, like in the *sc*Pmt1-Pmt2 structure[11], we did not observe clear density for a catalytic ion. Taken together, the structural similarities indicate that the catalytic mechanism is strictly conserved across PMT families.

Aside from the conserved interactions with the acceptor threonine residue, it was previously observed that *sc*Pmt2 does not specifically interact with the remainder of the bound PYT substrate peptide[11]. Regardless, the equivalent substrate binding pocket of *ct*Pmt4 does show some differences in peptide-proximal residues that could contribute to substrate recognition (Fig. 3C). Most notably, the *ct*Pmt4 δ′ site within the MIR linker does not contain the four residue insertion specific to the PMT2 family[14] and, as such, adopts a distinct conformation that brings it much closer to the expected position of the substrate peptide. However, the *ct*Pmt4 peptide-proximal residues and δ′ site are virtually identical to those within *sc*Pmt1 (Supplementary Fig. 8E). This suggests that these features are unlikely to be strong determinants of substrate specificity, as would be expected given PMTs do not recognise their substrates via defined sequence motifs[2]. Interestingly, the attached MIR domains in the *ct*Pmt4 homodimer and *sc*Pmt1-Pmt2 heterodimer occupy significantly different positions relative to the TMDs, with the *ct*Pmt4-MIR domain sitting much closer to the substrate peptide binding site (Fig. 3C). It is therefore tempting to speculate a role for the Pmt4-MIR domain in selecting membrane protein substrates[17], either through direct substrate recognition or by sterically favouring membrane protein access to the active site.

## Pmt4 binds Dol-P-Man on the cytosolic side of the membrane

The head group of a second single-chain lipid is bound within a positively charged pocket adjacent to the cytosolic end of TMD9 (Fig. 3A). To identify the lipid species, *ct*Pmt4 was purified in dodecyl maltoside (DDM) detergent (Supplementary Fig. 2A) and subjected to native mass spectrometry (native MS). The resulting spectrum confirms that *ct*Pmt4 also forms a homodimer in this detergent, and shows that both monomer and dimer are associated with adduct peaks corresponding to bound ~1100 Da and ~1700 Da species (Fig. 4A and Supplementary Table 1). Subsequent lipidomics analysis revealed that, in addition to a mixture of the common <1000 Da phospholipids present in the *S. cerevisiae* ER membrane (Supplementary Fig. 9A, B), >1000 Da Dol-P-Man and cardiolipin are associated with the purified protein (Fig. 4B and Supplementary Fig. 9C). Further dissociation of bound lipid species during native MS indicated that the ~1100 Da adduct peaks correspond to Dol-P-Man with an average of 13 isoprene units (Supplementary Fig. 9D). As observed previously[22], the mitochondrial lipid cardiolipin is likely present as an artefact of protein purification from total cellular membranes, suggesting that detergent extraction can to some extent alter native lipid composition. Intriguingly, cardiolipin (~1400 Da) is not large enough to fully account for the ~1700 Da adduct peaks observed in native MS; these are therefore likely to correspond to a detergent adduct with a lipid, for example, cardiolipin or the Dol-P present in the active site.

Strikingly, we found that Dol-P-Man is a good fit for the lipid density in the cytosolic binding pocket, with the phosphate group being stabilised by positively charged residues (R218, R638 and R642) and the mannose group by histidine side chains (H219 and H637) (Fig. 3D and Supplementary Fig. 8A). This coordination is highly analogous to the binding of Dol-P product, and ergo Dol-P-Man substrate, in the active site of PMT proteins (Fig. 3B and Supplementary Fig. 8C). Similarly, the density for the head group is weaker than for the dolichol

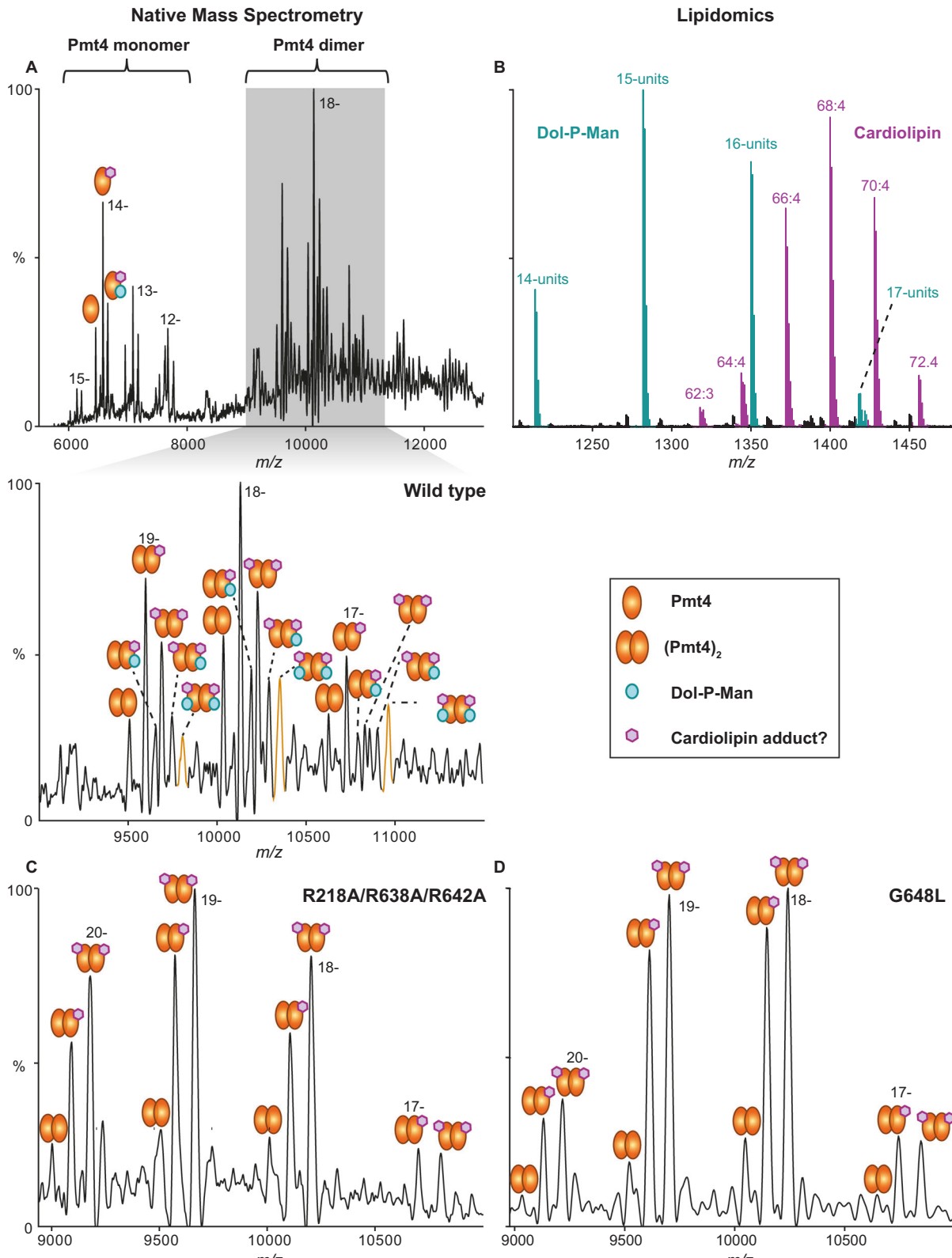

**Fig. 4 | Native mass spectrometry analysis of Dol-P-Man binding to *ct*Pmt4.**
**A** Mass spectrum of *ct*Pmt4 liberated from DDM micelles collected under non-denaturing conditions. For each assigned species, the charge state for the most intense peak is shown. The zoomed region shows the main charge states for dimeric *ct*Pmt4. Adduct peaks with a defined mass greater than the amino acid sequence mass correspond to a mixture of bound Dol-P-Man and a putative adduct between detergent and a lipid such as cardiolipin. Based on the relative abundance,

~40% *ct*Pmt4 dimers are bound to at least one molecule of Dol-P-Man. **B** MS profile of dolichol and cardiolipin species identified through in-solution lipidomics. The workflow for assigning Dol-P-Man as the major dolichol species is shown in more detail in Supplementary Fig. 9. Mass spectra of **C** R218A/R638A/R642A and **D** G648L mutants of *ct*Pmt4 in the region corresponding to the dimeric species. Adduct peaks corresponding to bound Dol-P-Man are no longer present. The full spectra are shown in Supplementary Fig. 10B, C, respectively.

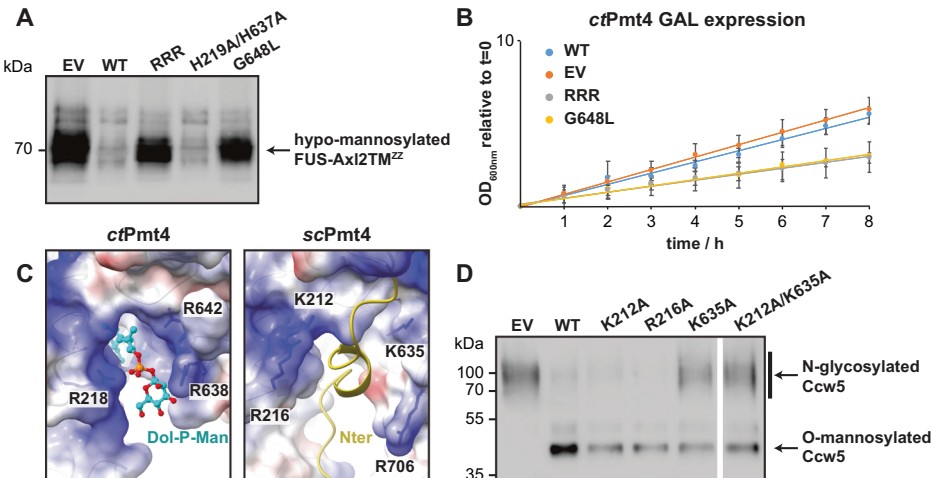

**Fig. 5 | Dol-P-Man binding to a cytosolic pocket is functionally important for Pmt4. A** Western blot analysis of the reporter protein FUS-Axl2TM$^{ZZ}$ isolated from yeast strain *pmt4-G/AXL* transformed with empty pREP3-adh vector (EV), wild-type *ct*Pmt4 (WT) or Dol-P-Man binding mutants under the control of the constitutive *ADH1* promoter. 40 µg total membrane protein from each strain was separated on an 8% SDS-PA gel and probed with peroxidase-coupled anti-rabbit-IgG antibody. A representative blot is shown from three independent experiments. **B** Growth of the yeast *pmt4-G* strain transformed with empty pMT929 vector (EV), wild-type *ct*Pmt4 (WT) or Dol-P-Man binding mutants. Selective medium was supplemented with 2% galactose to overexpress variants from a *GAL1/GAL10* promoter and OD$_{600nm}$ was measured every 60 min. Average values from three independent experiments are shown on a logarithmic scale with error bars showing the standard deviation. **C** Side-by-side comparison of the cytosolic pockets in *ct*Pmt4 and the *sc*Pmt4 AlphaFold DB[23,24] model after superimposition of their TMDs (RMSD 1.00 Å over 487 Cα-atoms). The electrostatic surface potentials (± 10 kT) are shown and positively charged residues are labelled. In *ct*Pmt4 the pocket is occupied by Dol-P-Man, whilst in *sc*Pmt4 it is occupied by its N-terminus (Nter), shown in cartoon representation for clarity. **D** Western blot of Ccw5 isolated from a yeast Δ*pmt4* strain expressing FLAG-tagged Ccw5 (*pmt4-G/CCW5*) transformed with wild-type *sc*Pmt4 (pJK4-B1; WT), empty pRS423 vector (EV) and pJK4-B1 with mutations in positively charged residues in the cytosolic pocket. Anti-FLAG antibody was used to detect O-mannosylated and N-glycosylated isoforms of Ccw5 separated on a 10% SDS-PA gel. A representative blot is shown from three independent experiments. Source data for **A**, **B** and **D** are provided in the Source Data file.

tail, implying some flexibility within the Dol-P-Man binding site. To further verify this Dol-P-Man interaction, alanine mutants of the coordinating arginine and histidine residues in the *ct*Pmt4 cytosolic pocket were purified and were confirmed using nanoDSF to have the same thermal stability in DDM as the wild-type protein (Supplementary Fig. 10A). Subsequently native MS showed that, whilst single arginine or double H219A/H637A mutations were insufficient to prevent Dol-P-Man binding (Supplementary Fig. 11), double R218A/R642A or triple R218A/R638A/R642A (RRR) mutations abolished the interaction (Fig. 4C and Supplementary Fig. 10), confirming that Dol-P-Man is the single-chain lipid species bound within the cytosolic pocket of *ct*Pmt4.

Only the first six isoprene units of Dol-P-Man are resolved in the density, whilst lipidomics analysis shows that species with up to 17 units are associated with *ct*Pmt4 (Fig. 4B and Supplementary Fig. 9C). This suggests that more than half of the dolichol tail is disordered within the central rhombic cavity, as for Dol-P within the active site. Of the ordered portion, the first two isoprene repeats are accommodated in a hydrophobic groove along the membrane normal between TMD4, TMD6 and TMD9 (Fig. 3A and Supplementary Fig. 8B). However, this trajectory is interrupted by the side chain F649, which sits in the groove and forces the dolichol tail to kink 120° (Fig. 3E). The remaining resolved isoprene repeats then adopt an alternative path along the membrane plane, which crosses hydrophobic surfaces of TMD8 and TMD9 (Fig. 3A, E and Supplementary Fig. 8B). In fact, the dolichol tail passes so close to G648 within TMD9 that we reasoned a larger side chain in this position would sterically impede Dol-P-Man binding. Indeed, native MS showed that Dol-P-Man was no longer bound to a G648L mutant of *ct*Pmt4, despite it exhibiting the same thermal stability as the wild-type protein (Fig. 4D and Supplementary Fig. 10A, C). Therefore, the Dol-P-Man interaction requires both stabilisation of its dolichol tail in the observed conformation and its phosphate-mannose headgroup in the cytosolic binding pocket.

## Dol-P-Man binding to a cytosolic pocket is functionally important for Pmt4

We next wanted to assess whether the observed binding of Dol-P-Man to the cytosolic pocket of *ct*Pmt4 is important for its function. First, the O-mannosyltransferase activity of *ct*Pmt4 in *S. cerevisiae* was demonstrated through complementation of the Δ*pmt1pmt4* mutant with *ct*Pmt4 expressed from a constitutive *ADH1* promoter. The in vivo functionality of *ct*Pmt4 was similar to *sc*Pmt4, restoring normal growth at elevated temperatures (Supplementary Fig. 12A) and alleviating the clumping phenotype observed in the *S. cerevisiae* Δ*pmt1pmt4* mutant (Supplementary Fig. 12B). HA-tagged *ct*Pmt4 (*ct*Pmt4-HA) was then shown to mannosylate the previously described chimeric *sc*Pmt4 substrate FUS-Axl2TM$^{ZZ}$ in Δ*pmt4* cells[17] (Fig. 5A), providing an assay system for in vivo *ct*Pmt4 activity. We verified the stability and membrane integration of RRR, G648L and H219A/H637A *ct*Pmt4-HA variants in the Δ*pmt4* strain (Supplementary Fig. 12C) and found that cells expressing RRR or G648L mutants of *ct*Pmt4-HA, which cannot bind Dol-P-Man at the cytosolic pocket, accumulated hypo-mannosylated FUS-Axl2TM$^{ZZ}$ at ~70 kDa compared to those expressing wild-type or H219A/H637A *ct*Pmt4-HA (Fig. 5A). Whilst constitutive expression of RRR and G648L *ct*Pmt4-HA did not impair the growth of Δ*pmt4* (Supplementary Fig. 12D), overexpression of these mutants from a *GAL* promoter had a dominant negative effect, leading to impaired cell growth (Fig. 5B) and increased cell clumping (Supplementary Fig. 12E) relative to overexpression of wild-type His-tagged protein (*ct*Pmt4-His). However, FUS-Axl2TM$^{ZZ}$ is hypo-mannosylated only in the presence of overexpressed G648L *ct*Pmt4-His, implying that increased protein amounts compensate for the mannosylation defect of the RRR *ct*Pmt4-His variant (Supplementary Fig. 12F). Nevertheless, altogether these results show that Dol-P-Man binding to the cytosolic pocket of *ct*Pmt4 impacts its O-mannosyltransferase activity.

We were intrigued whether cytosolic Dol-P-Man binding is a general feature important for protein O-mannosylation by PMTs, given

that residues involved in Dol-P-Man binding to *ct*Pmt4 are not strictly conserved (Supplementary Fig. 1). Strikingly, a comparison with the AlphaFold DB[23,24] or cryo-EM models for yeast and human PMT4 (*sc*Pmt4, *hs*POMT1), PMT1 (*sc*Pmt1) and PMT2 (*sc*Pmt2, *hs*POMT2) family members revealed a common positively charged cytosolic cavity in the same position as the Dol-P-Man binding site in *ct*Pmt4, particularly in the vicinity of the phosphate group (Fig. 5C and Supplementary Fig. 13A). Furthermore, a similar feature also appears to be present within the distinct O-mannosyltransferases TMTC1-4 and TMEM260 found in mammals[6] (Supplementary Fig. 13B). To assess whether this cavity is important for the enzymatic activity of *sc*Pmt4, alanine substitution mutants of the basic residues K212, R216 and K635 were expressed in a Δ*pmt4-CCW5* reporter strain. Whilst the stability and membrane integration of these variants in vivo was unchanged relative to wild-type *sc*Pmt4-FLAG (Supplementary Fig. 12G), expression of K635A and K212A/K635A *sc*Pmt4-FLAG lead to an increase in N-glycosylated Ccw5, demonstrating that these variants are impaired for protein O-mannosylation (Fig. 5D). These results provide a compelling argument for the conserved cytosolic cavity being functionally important at least within fungal Pmt4 and potentially beyond, although its role in binding Dol-P-Man needs to be verified for other PMTs. Interestingly, the cytosolic N-termini, which in our *ct*Pmt4 homodimer structure are not resolved and likely flexible, at least partially occupy the cytosolic cavity within *sc*Pmt1 and *sc*Pmt4 models (Fig. 5C and Supplementary Fig. 13A) and thus could act as an alternative natural ligand to Dol-P-Man.

## Discussion

Our structure of a fungal *ct*Pmt4 homodimer confirms that the high sequence conservation between the PMT1, PMT2 and PMT4 families (Supplementary Fig. 1) corresponds to highly similar protein folds for both their TMDs and MIR domains, and to the same overall arrangement of their obligate dimers within the membrane region[11]. The residue arrangement within the active site of PMTs and the mode of Dol-P product binding are invariant, implicating a shared inversion reaction mechanism for mannosyl transfer. Furthermore, the conservation of mannose-binding sites within Pmt4-MIR domains suggests they have a similar role in ensuring a processive enzyme reaction through the coordination of mannosylated peptide products[14].

Despite a conserved dimer interface within their TMDs, PMTs form mutually exclusive Pmt1-Pmt2 and Pmt4-Pmt4 dimer species, potentially with the help of lipids or additional family-specific binding elements. They also exhibit fundamental albeit broadly defined differences in substrate specificity (soluble vs membrane proteins, bona fide vs misfolded substrates), which can now begin to be rationalised on the basis of structural differences between the *ct*Pmt4 homodimer and *sc*Pmt1-Pmt2 heterodimer. A fundamental difference is the cis interaction between the MIR domains and TMDs in *ct*Pmt4 instead of the trans interaction in *sc*Pmt1-Pmt2, accompanied by a different arrangement of the poorly conserved flexible linker regions. As a result, the *ct*Pmt4-MIR domain sits virtually atop the TMDs in closer proximity to the luminal active site, compared to the *sc*Pmt1-Pmt2 interaction. Therefore, the substrate and product accessibility to the active site and MIR domain surface may be altered such that fungal Pmt4 achieves its more stringent selection of bona fide membrane protein substrates[17]. An interesting paradigm will be the cis or trans nature of the MIR interactions within the human POMT1-POMT2 dimer, which comprises members of both the PMT4 and PMT2 families.

In both *ct*Pmt4 and *sc*Pmt1-Pmt2 dimer structures, only one MIR domain is engaged with the TMDs and the interaction interface, which in each case involves the MIR δ site, is weak. Therefore, independent of the cis or trans nature of the interaction, it is possible that the MIR domains alternately and transiently interact with the TMDs to regulate catalysis and/or access to the active site. Given the overall similarity

between the TMDs of *ct*Pmt4 and *sc*Pmt1-Pmt2, substrate selectivity could alternatively be achieved through higher-order complex formation with other biogenesis factors. For example, the observed association of PMTs with the Sec61 translocon and oligosaccharyltransferase may allow them to already select for particular substrates during their translocation[25]. However, more detailed insights into the catalytic mechanism, substrate specificity and regulation of PMTs require characterisation of the Pmt4 homodimer bound to specific substrates, and more generally, the different conformations and putative higher order assemblies of PMTs.

PMT4-MIR domains also contain features that distinguish them from PMT2-MIR domains[14], including an open γ site, a distinct δ site and the unique β-ins. Together with the conserved α and β sites, we suggest these sites represent further mannose binding sites that could confer additional specificity to the recognition of mannosylated peptides by the PMT4 family. In particular, we find that β-ins is required for O-mannosylation of substrate proteins by *sc*Pmt4 in vivo. Interestingly, the POMT1 residue W486, equivalent to the functionally important β-ins-stabilising W496 in *sc*Pmt4, has been found to be mutated in patients with congenital muscular dystrophy[26] (Supplementary Fig. 5), indicating this feature is generally needed for the function of PMT4-MIR domains.

Our structure revealed an additional molecule of substrate Dol-P-Man bound to a cytosolic pocket within the TMDs of *ct*Pmt4, which was not anticipated from previous data. This positively charged binding pocket is required for O-mannosylation by fungal Pmt4, and appears to be a generally conserved feature amongst PMTs and perhaps other O-mannosyltransferases. Furthermore, the G648/F649 motif in *ct*Pmt4, important for binding the dolichol tail of Dol-P-Man, is homologous to a GA motif in POMT1, of which A641 has been found to be mutated in human brain tumours[27] (Supplementary Fig. 1). Altogether this suggests the binding of cytosolic Dol-P-Man is widely important, however the physiological role of the interaction can currently only be hypothesised. To this end, it is interesting that Dol-P-Man itself is synthesised in the cytosolic leaflet of the ER membrane by Dol-P-Man synthase[28]. In addition, the head groups of the cytosolic Dol-P-Man and the active site Dol-P product are bound at opposite ends of a continuous groove through the membrane formed by TMD4, TMD6 and TMD9 (Fig. 3A), suggesting a route for communication between the two sites. One possibility is that the cytoplasmic pocket is an allosteric site that senses the availability of Dol-P-Man, with our data suggesting binding of this lipid would positively regulate catalytic activity. Alternatively, Pmt4 could be a long sought after Dol-P-Man flippase[29], directly moving Dol-P-Man from the cytosolic binding site, through the TMD groove to the active site and thereby coupling flipping to catalysis. Indeed, within this TMD groove there is a highly conserved HYΦP motif in TMD9 that is identical to a motif involved in binding Dol-P(-Man) in the active site (Supplementary Fig. 1) and, as such, could provide an intermediate binding site for the lipid head-group as it is flipped. However, Dol-P-Man flipping would require a significant conformational change in *ct*Pmt4 for the dolichol tail to bypass the narrow constriction between TMD6 and TMD9 and the PMT structures to date have shown a rather invariant TMD arrangement (Fig. 1C). Therefore, this study scratches the surface of our understanding of PMTs and lays a foundation to investigate further topics, such as the conformational landscape of these enzymes, the determinants of substrate specificity and the significance of cytosolic Dol-P-Man binding within a cellular context.

### Note added after review

After review of this manuscript, a complementary study by Du and colleagues was published, presenting the structure of *S. cerevisiae* Pmt4 in complex with a substrate peptide[30]. The overall architecture of the Pmt4 homodimer is conserved, and it is shown that β-ins within the MIR domain (denoted MIR-1 motif loop) interacts with the substrate

peptide, corroborating our functional data showing that β-ins is important for substrate O-mannosylation in vivo.

## Methods

### Construct design

All plasmids used in this study are shown in Supplementary Table 2. For recombinant protein expression, wild-type full-length *ct*Pmt4 was cloned with a C-terminal 3C-H$_{10}$ tag downstream of the bidirectional *GAL1/GAL10* promoter in the pMT929 vector[22] (pMT929 *ct*Pmt4). The *sc*Pmt4-MIR domain (residues 330-540), *sc*Pmt4-MIR Δ362-368 variant and *ct*Pmt4-MIR domain (residues 337-537) were cloned into a pET24d vector (Novagen) with an N-terminal MKH$_6$PM tag. pMT929 *ct*Pmt4 point mutations were subsequently created using the Quikchange Lightning site-directed mutagenesis kit (Stratagene).

For in vivo functional studies of *sc*Pmt4, K212A, R216A, K635A and K212A/K635A point mutations were introduced into plasmid pJK4-B1[8] using Quikchange Lightning site-directed mutagenesis kit, taking care to culture all subsequent *E. coli* transformants at 20 °C. Recombinant PCR[31] was used to introduce Δ362-368, Y362A/D364A/R366A, P427A/P430A and W496A mutations into pJK4-B1, with the resulting DNA fragments being cloned into NcoI/EcoNI linearised plasmid via homologous recombination in yeast. For constitutive expression in yeast strains, *ct*Pmt1 and *ct*Pmt4 were amplified from cDNA and cloned with a C-terminal HA tag into the pREP3-adh vector digested with NotI/XhoI or NotI/BamHI, respectively. Point mutations were introduced into pREP3-adh *ct*Pmt4-HA using Quikchange Lightning site-directed mutagenesis. All constructs were routinely validated through sequence analysis.

### S. cerevisiae strain construction

All *S. cerevisiae* strains used in this study are shown in Supplementary Table 3. To create Δ*pmt4-CCW5*, strain pmt4::*TRP1*[32] was transformed[33] with plasmid pML78[25]. To create strain *pmt4-G*, the *PMT4* deletion cassette was amplified from strain pmt4::*TRP1*[32] and transformed into DS1-2b[34]. Transformants were selected on synthetic complete (SC) medium -Trp and verified by PCR analyses. Strain *pmt4-G* was subsequently transformed with plasmids pML78[25] and pJH1[17] to create the reporter strains *pmt4-G/CCW5* and *pmt4-G/AXL*, respectively.

### Growth of strains

For recombinant protein expression, pMT929 *ct*Pmt4 variants were transformed into DS1-2b[34] and plated on SC -Trp plates for 48 h, before replating on SC -Leu plates for 24 h. A stationary SC -Leu pre-culture containing raffinose as the carbon source was diluted 12.5-fold into YP medium containing 1% galactose and incubated at 30 °C for 24 h. The Pmt4-MIR domains were expressed in *E. coli* Rosetta2 (DE3) (Novagen) by autoinduction[35] overnight at 18°C. All cells were harvested by centrifugation and stored at −20°C until further use.

Yeast strain CFY3[8] was transformed with pREP3-adh plasmids (*sc*Pmt1-HA, *sc*Pmt4-HA, *ct*Pmt1-HA, *ct*Pmt4-HA) and grown at 25 °C in SC -Leu medium until OD$_{600nm}$ ~ 1. From these cultures, light micrographs were taken, then a total of 10$^5$ cells and 10-fold serial dilutions thereof were spotted onto YPD plates, which were incubated for 3 days at 25 °C and 35 °C, respectively. pREP3-adh *ct*Pmt4-HA variants or pMT929 *ct*Pmt4 variants were transformed into yeast strains *pmt4-G* or *pmt4-G/AXL* and selected on SC -Leu or SC -Leu/-Ura media respectively. Verified transformants were grown at 30 °C in SC selective medium supplemented with 2% glucose (pREP3-adh *ct*Pmt4-HA variants) or 2% galactose (pMT929 *ct*Pmt4 variants). For growth curves of *pmt4-G* strains, the OD$_{600nm}$ was measured at 60 min intervals and light micrographs were taken during logarithmic growth.

Yeast strains Δ*pmt1pmt4*, Δ*pmt4-CCW5* or *pmt4-G/CCW5* were transformed with pRS423 empty vector, pJK4-B1 plasmid or mutated variants and selected on SC -His medium. Three individual colonies of

each Δ*pmt1pmt4* transformant were grown at 25 °C and 37 °C on YPAD plates for 3 days.

### Preparation of cell extracts and western blotting

Whole cell extracts and crude membranes were prepared for analysis of Pmt4 expression as previously described[15]. Cell wall extracts were prepared for analysis of Ccw5 glycosylation as previously described[16]. The Pmt4 substrate protein FUS-Axl2TM$^{ZZ}$ was analysed according to the protocol from Hutzler and coworkers[17]. Western blot analysis was performed as detailed by Bausewein and coworkers[18]. Monoclonal mouse anti-FLAG M2 (#F3165, Sigma-Aldrich) and anti-HA (#2362, Cell Signalling) antibodies were used at a dilution of 1:5000 and ZZ tag peroxidase-coupled anti-rabbit-IgG antibody from goat (#A6154, Sigma-Aldrich) was used at a dilution of 1:10,000. GAPDH-directed antibodies (#A9521, Sigma-Aldrich) and Sec61-directed antibodies (gift from K. Römisch) derived from rabbit were used at a dilution of 1:5000 and 1:2500, respectively. As secondary antibodies, peroxidase-coupled anti-rabbit-IgG antibody from goat (#A6154, Sigma-Aldrich) and anti-mouse antibody from rabbit (#A9044, Sigma-Aldrich) were used at a dilution of 1:10000 and 1:5000 respectively.

### Protein purification

For the purification of full-length *ct*Pmt4 variants, *S. cerevisiae* cell pellets were lysed in 50 mM HEPES (pH 7.5), 500 mM NaCl, 10% (v/v) glycerol, 1 mM EDTA by beating with glass beads at 500 rpm for 3 × 4 min. The lysate was cleared by centrifugation at 1000 xg for 25 min and the cell membranes were harvested and washed once by centrifugation at 160,000 xg for 30 min. The membranes were solubilised for 30 min in solubilisation buffer (50 mM HEPES (pH 7.5), 500 mM NaCl) supplemented with either 0.5% (w/v) LMNG (Anatrace) or 0.5% (w/v) DDM (Anatrace) for cryo-EM or native MS applications, respectively. These detergents were present in all subsequent buffers at a concentration closer to the critical micelle concentration: either a 0.01% LMNG/0.001% (w/v) CHS mixture (Anatrace) for cryo-EM or 0.03% (w/v) DDM for native MS. Solubilised membranes were centrifuged at 160,000 x *g* for 30 min and the supernatant was incubated with Ni-NTA agarose (Qiagen) for 30 min at 4 °C. This resin was washed with 40 column volumes (CV) wash buffer (solubilisation buffer with 30 mM imidazole) and *ct*Pmt4 was eluted by on-column cleavage via incubation with 5 mg His$_6$-tagged HRV-3C Protease in 5 CV wash buffer for 1 h at 4 °C. Size exclusion chromatography (SEC) was performed with a Superdex 200 increase 10/30 GL column (Cytiva) equilibrated with 20 mM HEPES (pH 7.5), 200 mM NaCl. Protein-containing fractions were concentrated with a VivaSpin Protein Concentrator MWCO 100,000 (Cytiva) to 20 μM for native MS.

For the purification of Pmt4-MIR domains, *E. coli* cell pellets were lysed in lysis buffer (50 mM HEPES (pH 7.5), 300 mM NaCl, 30 mM imidazole) using a M-110L Microfluidiser (Microfluidics). The lysates were cleared by centrifugation at 48,000 x *g* for 25 min, and the supernatant was applied to a 1 ml HisTrap HP column (Cytiva). The column was washed with 20 CV 50 mM HEPES (pH 7.5), 1 M NaCl, 30 mM imidazole, followed by 5 CV lysis buffer. The MIR domains were eluted with 50 mM HEPES (pH 7.5), 300 mM NaCl, 250 mM imidazole and applied to a HiLoad 16/60 Superdex 75 pg (Cytiva) column equilibrated in SEC buffer. The SEC buffer for *sc*Pmt4-MIR was 50 mM MES (pH 6.0), 400 mM NaCl for x-ray crystallography and 20 mM HEPES (pH 7.5), 150 mM NaCl for nanoDSF, whilst the SEC buffer for *ct*Pmt4-MIR was 20 mM HEPES (pH 7.5), 150 mM NaCl. Protein-containing fractions were concentrated with a 10 kDa Amicon-Ultra centrifugal filter (Merck Millipore).

### X-ray structure determination

All crystallisation was performed via the sitting drop vapour diffusion technique at 18 °C. The *sc*Pmt4-MIR domain was concentrated to 20.4 mg/ml and used to set 400 nl drops in a 1:1 ratio with 0.1 M HEPES

(pH 7.5), 20% (w/v) PEG 8000. Crystals were cryo-protected by soaking in mother liquor supplemented with 20% (v/v) glycerol and data were collected at beamline ID30A-1 at the European Synchrotron Radiation Facility (ESRF, Grenoble). Data were integrated with XDS[36] and scaled with AIMLESS[37]. Molecular replacement was performed in Phaser-MR[38] using a single molecule of *Arabidopsis thaliana* SDF2-like protein (PDB 3MAL) as a search model. Coot[39] and phenix.refine[40] of the Phenix package (version 1.15.2)[41] were used for iterative model building and refinement cycles.

The *ct*Pmt4-MIR domain was concentrated to 40 mg/ml and used to set 300 nl drops in a 1:1 ratio with 1.4 M sodium acetate (pH 7.0), 0.1 M sodium cacodylate (pH 6.5). The crystal was cryo-protected by soaking in mother liquor supplemented with 20% (v/v) ethylene glycol and data were collected at beamline ID30A-3 at the ESRF. Data were integrated with XDS and scaled with AIMLESS and STARANISO within the autoPROC toolbox[42]. Molecular replacement was performed in Phaser-MR of the Phenix package using the *sc*Pmt4-MIR domain as a search model, then model building and refinement as for the *sc*Pmt4-MIR structure. Crystallographic data, refinement and model statistics are summarised in Table 2.

## Cryo-EM grid preparation and data collection

*ct*Pmt4 was used at 1.7 mg/ml after SEC without further concentration and 3 µl sample was applied to a glow-discharged holey carbon-coated grid (Quantifoil 300 mesh, Cu 1.2/1.3), before plunge-freezing in liquid ethane using a Vitrobot Mark IV (Thermo Fisher). Freezing was performed at 100% humidity and 6 °C with 15 s adsorption, 5 s blot time, 10 blot force and 1 s drain time and grids were stored in liquid $N_2$ until data collection. Data were collected at beamline CM01 at the ESRF on a 300 kV Titan Krios transmission electron microscope on a K2 detector (Gatan). 9879 images were collected at 0.827 Å/pixel and a magnification of 165,000, with a total dose of 42.7 e⁻/Å² over 40 frames. EPU (Thermo Fisher) was used to set up and monitor data collection.

## Image processing

Motion correction was carried out with dose-weighting using the Relion 3.1[43] implementation of MotionCor2[44] and CTF estimation of unweighted micrographs was performed via GCTF[45]. 2,199,173 particles were picked using WARP[46] and imported into cisTEM 1.0 beta[47] for 2D classification followed by 3D auto-refinement of the selected classes into a single class. These centred particles were imported into Relion 3.1 for all subsequent processing. 3D classification (with image alignment) and 3D auto-refinement were followed by another round of 3D classification (without image alignment) and 3D auto-refinement. For the map without applied symmetry (C1), Bayesian polishing, 3D auto-refinement and post-processing yielded a 3.4 Å reconstruction. With C2 symmetry applied, Bayesian polishing, 3D auto-refinement, 3D classification without image alignment, 3D auto-refinement and post-processing yielded a 3.2 Å reconstruction. Both reconstructions were low-pass filtered to 6.0 Å to yield more continuous density for flexible, low-resolution regions. Local density sharpening of the C1 map was performed with LocScale[48] in order to represent each region at its local resolution. The data processing pipeline is summarised in Supplementary Fig. 3A.

## Model building and refinement

A SCWRL homology model[49] for the *ct*Pmt4 TMDs was generated from the *sc*Pmt2 TMDs (PDB 6P25) and sequence alignment in Supplementary Fig. 1, then jiggle-fit[50] into one subunit within the C2 reconstruction in Coot. Loop and helical deviations were manually repositioned to fit the density and the L566-P571 loop was built within the C2 map low-pass filtered to 6.0 Å. Models and geometry restraint files for Dol-P and Dol-P-Man were generated using eLBOW[51] of the Phenix package (version 1.15.2) then also fitted into the C2 density within Coot (version 0.8.9.2). The complete resulting model for one *ct*Pmt4 subunit was

then duplicated and jiggle-fit into the density for the second subunit within the C2 reconstruction. Real-space refinement in Phenix[40] with secondary structure, geometry and C2 symmetry restraints produced the final C2 model for the *ct*Pmt4 TMDs (Table 1). Subsequently, this TMD model and two copies of the *ct*Pmt4-MIR domain structure (with ligand, water molecules and flexible termini removed) were jiggle-fit into their corresponding densities within the C1 reconstruction in Coot. Finally, the linkers connecting the MIR domains and TMDs were manually built and real-space refined in Coot within the C1 map low-pass filtered to 6.0 Å. This produced the final composite model for the *ct*Pmt4 homodimer, which was validated in Phenix[52] against the C1 reconstruction (Table 1). Structural figures were prepared in UCSF ChimeraX (version 1.10)[53] and PyMOL (version 2.5; Schrödinger, LLC).

## nanoDSF

Full-length *ct*Pmt4 variants purified in 20 mM HEPES (pH 7.5), 200 mM NaCl, 0.03% (w/v) DDM for native MS were diluted to 0.3 mg/ml and two repeats for each variant were loaded into Prometheus standard capillaries (NanoTemper Technologies). The Prometheus NT.48 (NanoTemper Technologies) was used to monitor the temperature-dependent change in tryptophan fluorescence at 330 nm and 350 nm emission wavelengths, with a 1.5 °C/min temperature gradient from 20 °C to 90 °C. The melting temperature ($T_m$) was determined by detecting the maximum of the first derivative of the fluorescence intensity ratio at 350 nm and 330 nm. *sc*Pmt4-MIR domain variants purified in 20 mM HEPES (pH 7.5), 150 mM NaCl were diluted to 1 mg/ml and three repeats for each variant were measured as for *ct*Pmt4, except the Prometheus Panta (NanoTemper Technologies) was used with a 1 °C/min temperature gradient from 25 °C to 90 °C.

## Native MS

Protein samples were buffer exchanged into 200 mM ammonium acetate (pH 8.0) supplemented with 2x CMC of the respective detergent using Micro Bio-Spin 6 columns (BioRad) prior to MS analyses. These samples were directly introduced into the mass spectrometer using gold-coated capillary needles prepared in-house[54]. Data were collected on a Q Exactive UHMR Hybrid Quadrupole-Orbitrap mass spectrometer (Thermo Fisher) in negative polarity. The instrument parameters used for data collection were: capillary voltage 0.9 kV, S-lens RF 100%, quadrupole selection from 400 to 20,000 *m/z* range. The HCD collision energy was 300 V for all acquired spectra, with the exception of the mass spectrum shown in Supplementary Fig. 10E where it was 200 V. Source fragmentation was 0 V for all acquired spectra, with the exception of the mass spectrum shown in Supplementary Fig. 9D where it was 150 V. In-source trapping was 250 V for *ct*Pmt4 wild-type mass spectrum in Fig. 4A and 300 V for all other *ct*Pmt4 spectra. The ion transfer optics in negative mode was set as follows: injection flatapole −5 V, inter-flatapole lens −4 V, bent flatapole −2 V, transfer multipole 0 V. The resolution of the instrument was 8,750 at *m/z* = 200 (transient time of 64 ms) for all acquired spectra, with the exception of the mass spectrum shown in Supplementary Fig. 9D where it was 140,000 at *m/z* = 200. Argon pressure in the HCD cell was maintained at approximately 5 × 10⁻¹⁰ mbar and source temperature was kept at 250 °C. The noise level was set at 3 rather than the default value of 4.64. Calibration of the instruments was performed using 10 mg/ml solution of caesium iodide in water. Where required, baseline subtraction was performed to achieve a better-quality mass spectrum. Data were analysed using the Xcalibur 3.0 (Thermo Scientific), NaViA[55] and UniDec[56] software packages.

## Lipidomics analysis

Co-purified lipids have been identified in a similar manner as described previously[57]. Briefly, proteins were digested with trypsin (1:20 units) for 15 h at 37 °C in a thermomixer (Eppendorf). The digest was lyophilized and re-dissolved in 68% solution A (ACN:$H_2O$ 60:40, 10 mM

ammonium formate and 0.1% formic acid) and 32% solution B (IPA:ACN 90:10, 10 mM ammonium formate and 0.1% formic acid) for analysis by reverse phase liquid chromatography tandem MS (RP LC-MS/MS). LC-MS/MS was performed using a Dionex UltiMate 3000 RSLC Nano system coupled to an LTQ Orbitrap XL hybrid mass spectrometer (Thermo Fisher). The tryptic digest mixture was loaded onto a pre-equilibrated C18 trap (Acclaim PepMap m-Precolumn 300 mm, 5 mm) followed by a C18 column (Acclaim PepMap 100, C18, 75 mm 3 15 cm; Thermo Fisher) and separated by using the following step gradient of solution A and B at 300 nl/min and 40 °C: 0-10 min with 32% solvent B, solvent B was ramped to 65% over 1 min, then the concentration of solvent B was increased to 80% buffer over 6 min and held for 10 min, then ramped to 99% B over 6 min and this concentration was held for 7 min. The nanoflow reversed-phase liquid chromatography was directly coupled to the mass spectrometer via a dynamic nanospray source. Typical MS conditions were spray voltage of 1.6 kV and capillary temperature of 275 °C. The LTQ-Orbitrap XL was set-up in negative ion mode and in data-dependent acquisition mode to perform five MS/MS scans perMSscan. Survey full-scan MS spectra were acquired in the Orbitrap ($m/z$ 350 – 2000) with a resolution of 60,000 at $m/z$ = 200. Collision-induced dissociation (CID) fragmentation in the linear ion trap was performed for the five most intense ions at an automatic gain control target of 30,000 and a normalized collision energy of 38% at an activation of q = 0.25 and an activation time of 30 ms. Data were analyzed using Xcalibur 3.0 (Thermo Fisher).

## Reporting summary

Further information on research design is available in the Nature Portfolio Reporting Summary linked to this article.

## Data availability

Coordinates for the *sc*Pmt4 and *ct*Pmt4-MIR domains have been deposited in the Protein Data Bank (PDB) under accession numbers 9FD0 and 9FD1, respectively. The C2 and C1 cryo-EM volumes for *ct*Pmt4 have been deposited in the Electron Microscopy Data Bank under accession codes EMD-52631 and EMD-52632 respectively, and the C2 and C1 models in the PDB under accession numbers 9I5K and 9I5L respectively. PDB 3MAL and 6P25 were used to assist structure determination and model building. The native MS and lipidomics data have been deposited to the ProteomeXchange Consortium via the PRIDE repository with the dataset identifier PXD061837. Source data are provided with this paper.

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

## Acknowledgements

We thank Gunter Stier for the *scPmt4-MIR* domain construct. We also acknowledge Marvin Weiler for his assistance in purifying the *ct*Pmt4-MIR domain and Andreas Marz for mutagenesis of pJK4-B1. We thank Jürgen Kopp and Claudia Siegmann from the BZH/Cluster of Excellence: CellNetworks crystallisation platform and acknowledge access to beamlines ID30A-1, ID30A-3 and CM01 at the ESRF in Grenoble and the support of the beamline scientists. We acknowledge access to the infrastructure of the Cryo-EM Network at Heidelberg University (HDcryoNET). We also acknowledge the services SDS@hd and bwHPC supported by the Ministry of Science, Research and the Arts Baden-Württemberg (MWK) and the Deutsche Forschungsgemeinschaft (DFG; German Research Foundation) through grants INST 35/1314-1 FUGG and INST 35/1134-1 FUGG. This work was supported by the DFG through the Leibniz Programme (SI 586/6-1), FOR2509 (SI 586/8-2) and SFB1638 (P04) to I.S. and FOR2509 (STR 443/6-2) to S.S. C.V.R. is funded by the Wellcome Trust (221795/Z/20/Z) and F.F. by SAPIExcellence "TACKLE" (Sapienza University of Rome).

## Author contributions

M.A.M., K.W. and I.S. conceived the study and interpreted the results. M.A.M., A.C. and S.M. optimised and purified full-length Pmt4 samples for cryo-EM and native MS. M.A.M. performed nanoDSF on full-length Pmt4, collected EM data, processed EM data and built structural models. M.A.M. and Y.H. performed purification of MIR domain samples for crystallisation, crystallographic structure determination and model building. F.L.B. purified MIR domain samples for nanoDSF. K.W. performed refinement and analysis of structural models. F.F., D.W. and C.V.R. performed and analysed native MS and lipidomics experiments. D.Ba., D.Br., A.M. and S.S. performed and analysed functional assays in yeast. M.A.M., K.W. and I.S. wrote the manuscript. All authors contributed to the final version of the manuscript.

## Funding

## Competing interests
