## [Transparent Peer Review file · Nature Communications]

Structural characterisation of the fungal Pmt4 homodimer

Corresponding Author: Professor Irmgard Sinning

Version 0:

Reviewer comments:

Reviewer #1

(Remarks to the Author)

The manuscript describes the structural and biochemical characterization of fungal Pmt4. The authors determined the homodimeric enzyme's structure by cryo-EM and its poorly defined in cryo-EM maps MIR domain by X-ray crystallography. The overall structure is similar to a previously solved structure of a heterocomplex Ptm1-Ptm2 from the same family. The new observations include cis rather than in trans contacts between MIR and trans-membrane domain, differences in the structure of MIR domain, specifically the presence of -hairpin insertion, and observation of plausible cytoplasmic Dol-P-Man binding site. Using native mass spectroscopy, it is shown that Dol-P-Man co-purified with wild-type Pmt4 is depleted in mutants.

The important questions related to the difference in specificity are not addressed in the study, which makes it somewhat less impactful. Also, I could not find sufficient information in the manuscript about the catalytic site and substrate-binding site. Their detailed comparison between Pmt1-Pmt2 and Ptm4 would be a valuable addition to the manuscript.

Some mutagenesis studies require better controls. The mutation of the plausible Dol-P-Man binding site includes multiple positively charged residue substitutions by alanine residues. Do the mutations preserve the correct insertion of Ptm4 into the membrane? Are the mutants folded correctly?

Inspection of the cryo-EM density shows that Dol-P and Dol-P-Man densities can be modeled as such but there are ambiguities. The density modeled as Dol-P is fragmented. Coordination of the phosphate group by positively charged residues should have made its density well defined, yet, the density of strong electron scattering phosphate is weaker than that of the isoprenoid chain. This ambiguity must be explicitly mentioned.

On the cytoplasmic side, similarly, the density of Dol-P-Man is not well defined and the sugar ring only partially fits the density. The cryo-EM-based interpretation of the density as Dol-P-Man is plausible but not certain. Again, the phosphate group is not well defined. Given that the density is fragmented, it could also be interpreted as a lipid. The reconstruction shows densities for other aliphatic chains that are not bridged to allow interpreting them as lipids.

The study provides new experimental insights into Ptm4 structure and function but at the same time, leaves an impression of missing important fundamental insights into the subtype-specific substrate selection and catalytic mechanism.

In the discussion section, a plausible flippase function of Pmt4 is proposed. Do structural features support this suggestion? Is there any indication that Pmt4 may facilitate the negatively charged phosphate group of Dol-P-Man crossing the membrane?

Below are several additional specific comments:

(1) Page numbers are missing.

(2) What is the identity and homology between Pmt family members?

(3) From line 161. How do you know if the mutated MIR domain remained folded? The anti-FLAG western blots indicate the expression of the protein but not its correct insertion or folding.

Line 197: It would be helpful if the active site and substrate binding site could be explicitly shown in one of the main figures.

Line 201. I could not understand what it means: "... the active site is completely triangulated ..."

Line 217: All the aliphatic chain densities resolved in the map appear to be single-channel lipids. How are densities modeled as Dol-P or Dol-P-Man different from the others?

Table 1: Model resolution should be reported at 0.5 FSC cutoff. Cutoff 0.143 is used for two independently refined half-maps merged into one consensus map.

Reviewer #2

(Remarks to the Author)

The authors have combined crystallography, cryo-EM, native MS, and lipidomics to study the structure and composition of the Pmt4 homodimer and compare this to the known structure of the Pmt1-Pmt2 heterodimer. They also used mutagenesis, in particular to study the effect of specific mutations on the capacity of the dimer to bind Dol-P-Man. In my review, I focused on the mass spectrometry, as this most aligns with my own expertise. Overall, the MS work is solid, although I have a few suggestions that could potentially increase the level of confidence in some of the identifications.

*In native MS, particularly of large proteins and complexes, peak broadening due to adduct formation and incomplete desolvation is typical, and this is observed in Figure 4 as expected. However, this makes it difficult to identify small bound species such as lipids just from the mass difference between the bound and unbound state of the protein. Indeed, the authors write on line 243 that the mass differences are ~1100 Da and ~1700 Da, and in Table S1, the observed mass increases due to cardiolipin or Dol-P-Man binding fluctuate between proteins. The authors also state on lines 249-250 that the ~1700 Da mass increase cannot be due to cardiolipin alone, but speculate that this might be due to simultaneous binding of a lipid and a detergent molecule (indeed, they refer to this assignment as "putative" several times). The identification of cardiolipin and Dol-P-Man in the lipidomics experiments lends credibility to the authors' assignments. However, given the importance of this result, I wonder whether the authors could consider performing a tandem MS experiment to dissociate the bound species from the dimer and measure their accurate mass in the low-mass region of the spectrum, and potentially even perform further characterization using MSn. This approach has been described by one of the groups involved in the current work; see for example Gault et al., Nat Methods, 2020, or Wu et al., Nat Protoc, 2025.

*On lines 259-260, the authors write that "single arginine or double H219A/H637A mutations were insufficient to prevent Dol-P-Man binding" and refer to Figure S10. With the aforementioned caveat about the lipid identification, this statement seems fine. However, in the figure caption, they write twice that these mutations "do not affect binding" of the lipid. This seems like a much stronger statement to me and implies that there is no effect on affinity. I suggest moderating the figure caption and using a similar phrasing to that in the main text.

*In the native MS methods, the HCD energy range is given as 150-300 V, and source fragmentation as 0-100 V. These are significant ranges, so could the authors specify which spectra were acquired with which parameters?

*Also in the native MS methods, on line 676, the authors describe settings for positive mode; however, the native spectra in Figures 4, S9, and S10 all seem to have been acquired in negative mode. Can the authors clarify what work was carried out in positive mode, or is there a typo?

Minor points:

*Line 232: "we do not observed"  "we did not observe"

*Line 261: "R218/R638/R642"  "R218A/R638A/R642A"

*There is some artifact in the PDF file (possibly due to file conversion) affecting the phrases "GAPDH-directed antibodies" (starting on line 435) and "prepared in UCSF Chimera" (starting on line 612).

**"m/z" should be italicized

*Line 703: Can the authors specify the m/z value at which the resolution was defined?

*Data availability statement: The authors have uploaded their MS data to the PRIDE repository (and should be commended for this) so the dataset identifier should be added here.

Reviewer #3

(Remarks to the Author)

McDowell et al. provide the first structural insight into PMT4 homodimers which act as biosynthetic enzymes for fungal protein O-mannosylation. The authors have resolved the structure of *Chaetomium thermophilum* Protein O-mannosyltransferase 4 (Pmt4) homodimers by single-particle cryo-electron microscopy (CryoEM) and resolved high-resolution crystal structures of the fungal PMT4-MIR domains that protrude into the ER-lumen. Using native mass

spectrometry and lipidomic analysis, the authors provide further insight into PTM4 homodimer assembly and lipid composition to identify co-purified Dol-P-Man donor substrate. Guided by their structures, the authors map key residues in the PMT4 homodimer and PMT4-MIR domains which are further validated by mutations and deletions in functional studies. Remarkably, the authors identify two distinct sites with bound Dol-p and Dol-p-Man in the PMT4 homodimer, which opens for new hypotheses on how GT-C type enzymes may utilize donor substrates in catalytic cycles. These findings are truly novel and exciting as they potentially suggest an intrinsic “flippase” mechanism for positioning Dol-p-Man from the cytosolic side towards the ER-lumen. Further experiments and structural studies are clearly needed to test this hypothesis; however these findings will likely change current views and have boarder impact on the understanding of the mammalian POMT1/POMT2 orthologs, and by extension, also other related enzymes (DPY19L1-4, TMTC1-4 and TMEM260). Overall, the manuscript is well written, the experiments skillfully executed, the results are detailed and clear. The discussion is balanced and aligned with the results, and the authors provide insightful conclusions. I have minor comments and suggestions for the authors (below) but my enthusiasm for the work is high and I would argue that the manuscript should be considered favorably.

General suggestions:

Line 27: Consider rephrasing the sentence. Reading the abstract, I misinterpreted the sentence to believe that CryoEM was resolved for both *Chaetomium thermophilum* and *Saccharomyces cerevisiae* Pmt4 homodimers.

Line 41: In addition to POMTs, mammalian O-mannosylation is also initiated by TMTCs and the recently described TMEM260 enzyme. Moreover, transfer of mannose to distinct protein substrates also includes the actions of DPY19Ls (C-Man), and all these enzymes are collectively classified as GT-C enzymes together with PMTs. The readership may benefit if this was clarified in the introduction to provide a broader scope/justification for this study.

Line 88: The introduction or result section may benefit from a paragraph explaining the evolutionary relationship between ctPmt4 and scPmt4, with a brief statement on why the ctPmt4 dimer was chosen for CryoEM studies.

Line 109: Are the TMD6/8 residues in the dimer interface in Pmt1-2 and Pmt4 present in other Pmts? Considering that Pmts don't randomly dimerize, does the Pmt4 dimer interface provide insight into the predominant dimerization events among Pmts? Is this a general feature also conserved in mammalian POMT dimers? These questions can enrich the discussion paragraph.

Line 142: Regardless of outcome, it would be useful to know if the authors have attempted to capture a potential interaction between scPmt4-MIR and O-mannosylated peptides, either in co-crystallization experiments or other biochemical assays. As is now, the statement on mannose binding/CBMs is rather speculative.

Line 190-196: Without experimental evidence for the “substrate/product peptide route”, this paragraph is rather speculative. I would suggest that the authors consider rephrasing “highly suggestive” with softer language.

Line 240: DDM extraction may, or may not, preserve the native lipid content of the assembly. This should be briefly stated for the readership.

Line 244: Why was the identification of cardiolipin interesting? Was this expected or unexpected?

Line 287-300: The result paragraph for the functional assay with the FUS-Axl2TMZZ substrate is not perfectly clear to me. What was the rational? How many technical/biological replicates were done for Figure 5A? Why is there no loading control? These questions are rooted in my difficulties in interpreting the results in Figure 5A. If the assay based on assessing the absence/accumulation of the indicated band at ~68 kDa, why is there a difference in intensity between EV and WT and why is there no major accumulation in RRR and H219A/H637A (compared to EV)? The authors should consider clarifying.

Line 314-315: Does this argument also apply to other GT-C enzymes, e.g. DPY19Ls or TMTCs?

Line 320: The discussion should perhaps briefly state that one current limitation is lack of insight into the Pmt4 dimer-substrate peptide complex, which precludes further insight into the catalytic cycle of the reaction.

Line 1045: Figure S1: Size exclusion elution profiles of ctPmt4 appear to migrate as monomers? It may be beneficial to indicate the migration pattern of molecular standards on the Superdex 200 Increase SEC. It may be further beneficial to indicate which elution fractions were used for CryoEM. It is unclear why ctPmt4 migrates at ~100 kDa (LMNG-CHS) and ~70 kDa (DDM) in the following SDS-PAGE analysis?

Version 1:

Reviewer comments:

Reviewer #1

(Remarks to the Author)

The authors addressed all my comments adequately and the manuscript can be published.

Reviewer #3

(Remarks to the Author)

The authors have addressed my comments.

Response to reviewers

First of all, we would like to thank all reviewers for their thoughtful comments and constructive criticism, which have helped improve the quality of our manuscript.

Reviewer #1 (Remarks to the Author):

The manuscript describes the structural and biochemical characterization of fungal Pmt4. The authors determined the homodimeric enzyme's structure by cryo-EM and its poorly defined in cryo-EM maps MIR domain by X-ray crystallography. The overall structure is similar to a previously solved structure of a heterocomplex Ptm1-Ptm2 from the same family. The new observations include cis rather than in trans contacts between MIR and trans-membrane domain, differences in the structure of MIR domain, specifically the presence of β -hairpin insertion, and observation of plausible cytoplasmic Dol-P-Man binding site. Using native mass spectroscopy, it is shown that Dol-P-Man co-purified with wild-type Pmt4 is depleted in mutants.

The important questions related to the difference in specificity are not addressed in the study, which makes it somewhat less impactful. Also, I could not find sufficient information in the manuscript about the catalytic site and substrate-binding site. Their detailed comparison between Pmt1-Pmt2 and Ptm4 would be a valuable addition to the manuscript.

Although Pmt1-Pmt2 and Pmt4 act on distinct proteins, the determinants of their substrate specificity are very poorly defined in the literature. Indeed, there are no strict sequence motifs for substrate recognition, rather a vague propensity for Ser/Thr rich regions in hydrophilic, intrinsically disordered domains (Neubert et al., 2016). In terms of specificity, Pmt1-Pmt2 acts on soluble and membrane proteins, whilst Pmt4 seems to preferentially mannosylate membrane proteins (Hutzler et al., 2007). Some substrates (e.g. CCW5, Ecker et al., 2003) are mannosylated by both Pmt1-Pmt2 and Pmt4. Altogether, the substrate specificity is broad, overlapping and ill-defined. Therefore, it is to be expected that the structures do not show clear-cut differences that can be easily correlated with specificity, rather actually there is likely to be quite some overlap in their features.

In the manuscript, we compare the active site of Pmt4 with those of Pmt1 and Pmt2 (Figures 3B and S7C) and find no difference in the residue arrangement, aside from the 60° rotation in R666 that we describe in the text. We therefore conclude that the catalytic mechanism is conserved across the PMT families and, as we cannot extend mechanistic insights beyond Bai et al., 2019, do not discuss this in much more detail.

To rationalize differences in substrate specificity in terms of structure, we rely on differences between the structures of Pmt1-Pmt2 and the Pmt4 homodimer, with the caveat that our current structure does not contain bound substrate peptide. Within the TMDs, these structures have many more similarities than they have differences, whilst in the Pmt1-Pmt2 structure, the only specific contacts between Pmt2 and the bound PYT peptide are with the acceptor threonine residue (Bai et al., 2019). Nevertheless, we now add a comparison of the equivalent substrate binding region in Pmt4, Pmt1 and Pmt2 (Figure 3C and S7E). We find some differences in the peptide-proximal residues and δ' site between Pmt4 and Pmt2, however these features are identical in Pmt1 indicating they do not hugely contribute to differences in substrate specificity between these families. Another notable difference is in the positioning of the attached MIR domains with respect to the TMDs, where the Pmt4 MIR domain sits much closer to the active site and could thereby also contribute to substrate selection. We have now added these observations to the results section.

Given that there is some evidence that PMTs are found in a complex with the Sec61 translocon and OST (Loibl et al., 2014), it could also be that Pmt1/Pmt2 and Pmt4 achieve substrate specificity by interacting with different translocon-associated complexes e.g. in response to the translocation of particular substrates. This more speculative point has been added to the discussion.

Some mutagenesis studies require better controls. The mutation of the plausible Dol-P-Man binding site includes multiple positively charged residue substitutions by alanine residues. Do the

mutations preserve the correct insertion of Ptm4 into the membrane? Are the mutants folded correctly?

All ctPmt4 mutants used for both native MS and *in vivo* assays were purified from membranes and subjected to nanoDSF, showing no change in their unfolding temperature relative to the wild type protein (Figure S9A). This strongly indicates that the wild type and mutant ctPmt4 variants have the same fold. This is further supported by native MS, which shows that dimer formation is not disrupted and that the putative cardiolipin adduct is still able to bind (Figure 4). These mutants were also present at equivalent levels to the wild type protein in the membrane (Figure S11C). We assert that together this already provides sufficient evidence that the mutants are correctly inserted in the membrane and properly folded.

Unfortunately, it is not possible to purify the equivalent mutants of scPmt4 in sufficient amount and purity for the same analysis. However, these are also present at the same level as the wild type protein in the membrane (Figure S11G), indicating correct insertion and, as incorrect folding of the TMDs would likely lead to impaired membrane integration or membrane extraction, likely correct folding.

Inspection of the cryo-EM density shows that Dol-P and Dol-P-Man densities can be modeled as such but there are ambiguities. The density modeled as Dol-P is fragmented. Coordination of the phosphate group by positively charged residues should have made its density well defined, yet, the density of strong electron scattering phosphate is weaker than that of the isoprenoid chain. This ambiguity must be explicitly mentioned.

On the cytoplasmic side, similarly, the density of Dol-P-Man is not well defined and the sugar ring only partially fits the density. The cryo-EM-based interpretation of the density as Dol-P-Man is plausible but not certain. Again, the phosphate group is not well defined. Given that the density is fragmented, it could also be interpreted as a lipid. The reconstruction shows densities for other aliphatic chains that are not bridged to allow interpreting them as lipids.

We agree with the reviewer that the densities for Dol-P and Dol-P-Man are not as well resolved as for the surrounding TMDs. In both cases, this is likely due to two reasons:

- 1. The headgroups exhibit flexibility with respect to the isoprenoid chain, explaining why the densities for the phosphate groups are weaker than expected.**
- 2. These lipids only have partial occupancy in the structure. From the relative abundance of the species resolved in the native mass spectrum in Figure 4A, we can estimate that ~40% Pmt4 dimers have at least one molecule of Dol-P-Man bound. Dol-P species are not unambiguously resolved by native MS, so their occupancy is likely to be even lower, in-line with the weaker, more fragmented density relative to Dol-P-Man.**

We have added some clarification of these points in the text.

The modelling of the active site density as Dol-P is in full agreement with the binding and coordination of this species previously observed in the Pmt1-Pmt2 structure (Bai et al., 2019). Indeed, inspection of the Dol-P density in the Pmt1-Pmt2 map confirms that the density for the phosphate group here is also weaker than for the isoprenoid chain. The phosphate group occupies a slightly different position compared to the Pmt1-Pmt2 model, which is supported by the clear density of the rotated R666 side chain. The repositioning of Dol-P adds further weight to the argument that the headgroup has some flexibility within the active site; indeed, as the product of the reaction one would expect that the lipid should not be stably bound.

The modelling of Dol-P-Man in the cytosolic site is of course novel and required more careful justification. Our rationale was as follows: 1) the density unambiguously corresponds to a single chain and is not fragmented. 2) the coordination of the headgroup by His and Arg residues closely mimics the binding of Dol-P-(Man) in the active site. The sugar likely has some flexibility given that the capping cytoplasmic loop (residues 695-728) is also not resolved. 3) we clearly see Dol-P-Man in the native MS and do not see density to assign Dol-P-Man in the active site. To further verify the cytoplasmic lipid is Dol-P-Man, we made mutants in the binding site and confirmed they are still membrane integrated and folded (see response to previous point). Dol-P-Man

binding was abolished in several of these mutants (but notably not binding of the cardiolipin adduct), which in our view confirms our modelling of Dol-P-Man.

The study provides new experimental insights into Ptm4 structure and function but at the same time, leaves an impression of missing important fundamental insights into the subtype-specific substrate selection and catalytic mechanism.

Substrate specificity has been discussed more extensively in response to the first point, but we have added Figure 3C, Figure S7E and corresponding text to address this. With regards to the catalytic mechanism, the Pmt4 and Pmt1-Pmt2 catalytic sites are virtually identical, strongly indicating a conserved "inversion reaction mechanism for mannosyl transfer". As we do not resolve a bound metal ion, substrate peptide or Dol-P-Man in the active site, we cannot extend the description of the catalytic mechanism beyond what is already presented in Bai et al., 2019. Therefore, we focused our manuscript more extensively on the novel features of Pmt4 with respect to Pmt1-Pmt2.

In the discussion section, a plausible flippase function of Pmt4 is proposed. Do structural features support this suggestion? Is there any indication that Pmt4 may facilitate the negatively charged phosphate group of Dol-P-Man crossing the membrane?

H657 points into the TMD groove connecting the cytosolic and luminal Dol-P-Man binding sites and is part of a highly conserved HYΦP motif in TMD9. Interestingly, this motif is also found in the active site, where H672 interacts with the Dol-P(-Man) headgroup. We therefore think this motif forms a plausible intermediate binding site for the lipid headgroup as it is flipped. We have now added this point to the discussion and indicate the HYΦP motif below the sequence alignment in Figure S1.

Below are several additional specific comments:

(1) Page numbers are missing.

We apologise for this omission – we believe there was an error during the PDF conversion.

(2) What is the identity and homology between Pmt family members?

This is variable, but identity within a PMT family is typically >30% and between PMT families 25-30%. ctPmt4 and scPmt4 have a sequence identity of 48%. To illustrate this, a percent identity matrix including all *S. cerevisiae* homologues is now shown below the PMT sequence alignment (now Figure S1).

(3) From line 161. How do you know if the mutated MIR domain remained folded? The anti-FLAG western blots indicate the expression of the protein but not its correct insertion or folding.

The western blot in Figure S5B has now been obtained with samples derived from total membranes rather than whole cell lysate, demonstrating that all the MIR domain mutants are present in the membrane at the same level as the wild type protein. In addition, we created the most severe β -ins mutation, the β -ins deletion (Δ 362-368), in our scPmt4 MIR domain construct. We found that this variant could be purified to the same yield and homogeneity as the wild type construct, and that the domain was folded when assessed by nanoDSF, albeit with a slightly reduced stability. These points have been added to the text and the comparison of nanoDSF traces for the wild type and Δ 362-368 variants are shown as Figure S5A.

Line 197: It would be helpful if the active site and substrate binding site could be explicitly shown in one of the main figures.

The panel in Figure 3B shows the active site of ctPmt4 along with the position of dol-P and the acceptor threonine derived from the scPmt2 structure. It is superimposed with scPmt1 to demonstrate the highly conserved arrangement of residues in the active site. The equivalent superimposition with scPmt2 is shown in Figure S7C.

We have now added panels as Figure 3C and Figure S7E plus a corresponding description in the text, which show the substrate peptide binding sites of scPmt2 or scPmt1 respectively superimposed with ctPmt4.

Line 201. I could not understand what it means: "... the active site is completely triangulated ..."

We changed the wording to "the active site is completely covered by this δ site" to improve clarity.

Line 217: All the aliphatic chain densities resolved in the map appear to be single-channel lipids. How are densities modeled as Dol-P or Dol-P-Man different from the others?

We agree there are other densities, particularly in the C1 map, that could correspond to lipids, but they are generally more fragmented compared to the assigned Dol-P-Man density. We further cannot rule out that these additional densities also correspond to Dol-P species, but we have no evidence to assign them as such. Our rationale for modelling Dol-P and Dol-P-Man in their respective densities was primarily dependent on the chemistry of their binding sites and experimental validation, as outlined in more detail above in response to the previous point regarding these lipids.

Table 1: Model resolution should be reported at 0.5 FSC cutoff. Cutoff 0.143 is used for two independently refined half-maps merged into one consensus map.

Model resolution at 0.5 FSC cutoff are now given instead in Table 1.

Reviewer #2 (Remarks to the Author):

The authors have combined crystallography, cryo-EM, native MS, and lipidomics to study the structure and composition of the Pmt4 homodimer and compare this to the known structure of the Pmt1-Pmt2 heterodimer. They also used mutagenesis, in particular to study the effect of specific mutations on the capacity of the dimer to bind Dol-P-Man. In my review, I focused on the mass spectrometry, as this most aligns with my own expertise. Overall, the MS work is solid, although I have a few suggestions that could potentially increase the level of confidence in some of the identifications.

*In native MS, particularly of large proteins and complexes, peak broadening due to adduct formation and incomplete desolvation is typical, and this is observed in Figure 4 as expected. However, this makes it difficult to identify small bound species such as lipids just from the mass difference between the bound and unbound state of the protein. Indeed, the authors write on line 243 that the mass differences are ~ 1100 Da and ~ 1700 Da, and in Table S1, the observed mass increases due to cardiolipin or Dol-P-Man binding fluctuate between proteins. The authors also state on lines 249-250 that the ~ 1700 Da mass increase cannot be due to cardiolipin alone, but speculate that this might be due to simultaneous binding of a lipid and a detergent molecule (indeed, they refer to this assignment as "putative" several times). The identification of cardiolipin and Dol-P-Man in the lipidomics experiments lends credibility to the authors' assignments. However, given the importance of this result, I wonder whether the authors could consider performing a tandem MS experiment to dissociate the bound species from the dimer and measure their accurate mass in the low-mass region of the spectrum, and potentially even perform further characterization using MSn. This approach has been described by one of the groups involved in the current work; see for example Gault et al., Nat Methods, 2020, or Wu et al., Nat Protoc, 2025.

We thank the reviewer for this insightful suggestion. Indeed, we attempted to perform tandem MS experiments using both the Q Exactive UHMR (the instrument used for this study) and the Orbitrap Eclipse to dissociate and analyse the bound species in the lower m/z region. However, under our experimental conditions, the observation of the protein complex necessitates the application of collision voltages in the HCD cell downstream of the quadrupole for the Q Exactive UHMR. This step is essential because application of in-source trapping voltages could not effectively liberate the protein from the surrounding micelles. As a result, we are unable to isolate a micelle-free protein complex within the quadrupole for subsequent selection and activation steps.

This limitation precludes the isolation of a specific charge state of the protein-lipid complex prior to activation, which is a prerequisite for the type of tandem MS (and potential MSⁿ) analysis suggested by the reviewer. We fully agree that such an approach would offer valuable structural insights, and we are actively exploring alternative experimental strategies that might enable this type of analysis in future work. As noted, we also attempted native MS experiments using the Orbitrap Eclipse, but were unable to obtain sufficiently resolved spectra. The challenges encountered may stem from the heterogeneity of the protein system, which involves a monomer/dimer equilibrium and the presence of multiple bound lipid species, all contributing to a high degree of spectral complexity.

To further probe the identity of the bound species, we conducted an additional experiment by increasing the activation energies to the maximum levels permitted by the instrument (300 V HCD, 150 V source fragmentation, and 300 V in-source trapping) and increasing the resolution to 140,000 at $m/z = 200$. In parallel, we also optimised the mass spectrometry parameters to enhance transmission of low-mass ions, facilitating the detection of smaller dissociated species. Under these optimised conditions, we were able to dissociate the bound Dol-P-Man and observe its accurate mass in the low-mass region. The resulting peaks showed excellent agreement with the Dol-P-Man species identified in our lipidomics analyses, lending further support to our assignment. This experiment is now included as Figure S8D. Unfortunately, the 1700 Da species could not be resolved any further using the same approach.

*On lines 259-260, the authors write that "single arginine or double H219A/H637A mutations were insufficient to prevent Dol-P-Man binding" and refer to Figure S10. With the aforementioned caveat about the lipid identification, this statement seems fine. However, in the figure caption, they write twice that these mutations "do not affect binding" of the lipid. This seems like a much stronger statement to me and implies that there is no effect on affinity. I suggest moderating the figure caption and using a similar phrasing to that in the main text.

We thank the reviewer for pointing this out. We have now revised the figure caption in line with the suggestion, replacing the wording "do not affect binding" to "do not substantially impair binding" to better reflect the interpretation presented in the main text.

*In the native MS methods, the HCD energy range is given as 150-300 V, and source fragmentation as 0-100 V. These are significant ranges, so could the authors specify which spectra were acquired with which parameters?

*Also in the native MS methods, on line 676, the authors describe settings for positive mode; however, the native spectra in Figures 4, S9, and S10 all seem to have been acquired in negative mode. Can the authors clarify what work was carried out in positive mode, or is there a typo?

We thank the reviewer for highlighting these unclear points. We have now clarified in the Methods section the specific voltages used for each spectrum shown. Regarding the polarity, the reviewer is correct, this was a typo. All native MS spectra, including those in Figures 4, S9 and S10, were acquired in negative ion mode, and we have corrected the text accordingly.

Minor points:

*Line 232: "we do not observed"  "we did not observe"

Corrected.

*Line 261: "R218/R638/R642"  "R218A/R638A/R642A"

Corrected.

*There is some artifact in the PDF file (possibly due to file conversion) affecting the phrases "GAPDH-directed antibodies" (starting on line 435) and "prepared in UCSF Chimera" (starting on line 612).

We thank the reviewer for pointing out that. It seems to be an artifact from the PDF conversion during the submission process.

*"m/z" should be italicized

This has been corrected in the text and Figure 4.

*Line 703: Can the authors specify the m/z value at which the resolution was defined?

The resolution was defined at *m/z* = 200. This has now been included in the text.

*Data availability statement: The authors have uploaded their MS data to the PRIDE repository (and should be commended for this) so the dataset identifier should be added here.

We have now added the relevant statement along with the dataset identifier to the data availability statement.

Reviewer #3 (Remarks to the Author):

McDowell et al. provide the first structural insight into PMT4 homodimers which act as biosynthetic enzymes for fungal protein O-mannosylation. The authors have resolved the structure of *Chaetomium thermophilum* Protein O-mannosyltransferase 4 (Pmt4) homodimers by single-particle cryo-electron microscopy (CryoEM) and resolved high-resolution crystal structures of the fungal PMT4-MIR domains that protrude into the ER-lumen. Using native mass spectrometry and lipidomic analysis, the authors provide further insight into PMT4 homodimer assembly and lipid composition to identify co-purified Dol-P-Man donor substrate. Guided by their structures, the authors map key residues in the PMT4 homodimer and PMT4-MIR domains which are further validated by mutations and deletions in functional studies. Remarkably, the authors identify two distinct sites with bound Dol-p and Dol-p-Man in the PMT4 homodimer, which opens for new hypotheses on how GT-C type enzymes may utilize donor substrates in catalytic cycles. These findings are truly novel and exciting as they potentially suggest an intrinsic "flippase" mechanism for positioning Dol-p-Man from the cytosolic side towards the ER-lumen. Further experiments and structural studies are clearly needed to test this hypothesis; however these findings will likely change current views and have boarder impact on the understanding of the mammalian POMT1/POMT2 orthologs, and by extension, also other related enzymes (DPY19L1-4, TMTC1-4 and TMEM260).

Overall, the manuscript is well written, the experiments skillfully executed, the results are detailed and clear. The discussion is balanced and aligned with the results, and the authors provide insightful conclusions. I have minor comments and suggestions for the authors (below) but my enthusiasm for the work is high and I would argue that the manuscript should be considered favorably.

We thank the reviewer for these positive comments and have addressed their specific suggestions below.

General suggestions:

Line 27: Consider rephrasing the sentence. Reading the abstract, I misinterpreted the sentence to believe that CryoEM was resolved for both *Chaetomium thermophilum* and *Saccharomyces cerevisiae* Pmt4 homodimers.

We have rephrased the abstract.

Line 41: In addition to POMTs, mammalian O-mannosylation is also initiated by TMTCs and the recently described TMEM260 enzyme. Moreover, transfer of mannose to distinct protein substrates also includes the actions of DPY19Ls (C-Man), and all these enzymes are collectively classified as GT-C enzymes together with PMTs. The readership may benefit if this was clarified in the introduction to provide a broader scope/justification for this study.

We thank the reviewer for this suggestion. The introduction has now been updated to put the POMTs in the context of these other enzymes involved in protein mannosylation.

Line 88: The introduction or result section may benefit from a paragraph explaining the

evolutionary relationship between ctPmt4 and scPmt4, with a brief statement on why the ctPmt4 dimer was chosen for CryoEM studies.

In our hands, scPmt4 has proved largely recalcitrant to purification and structural studies, whilst generally the recombinant production of *C. thermophilum* thermostable proteins has consistently yielded high resolution structures. We now start the results section with “to investigate the structure of Pmt4, we took advantage of its thermostable homologue from the thermophilic fungus *Chaetomium thermophilum*, which has 48 % sequence identity to yeast Pmt4” and include a percent identity matrix with the PMT sequence alignment (now Figure S1).

Line 109: Are the TMD6/8 residues in the dimer interface in Pmt1-2 and Pmt4 present in other Pmts? Considering that Pmts don't randomly dimerize, does the Pmt4 dimer interface provide insight into the predominant dimerization events among Pmts? Is this a general feature also conserved in mammalian POMT dimers? These questions can enrich the discussion paragraph.

This is an intriguing question, as the residues at the TMD6/8 dimer interface are relatively few and largely conserved across PMTs and POMTs (Figure S1), therefore don't provide a satisfactory explanation for the formation of distinct dimer species. Similarly, the TMD9 residues at the luminal constriction point in the Pmt4 dimer are also conserved. In the Pmt1-Pmt2 structure, the Pmt1 N-terminus, the Pmt1 MIR domain and a lipid additionally contribute to the dimer interface and could promote preferential heterodimer formation by these species. We have added this point to the text. This is presumably less important for POMTs where the formation of only one dimer species is possible.

Line 142: Regardless of outcome, it would be useful to know if the authors have attempted to capture a potential interaction between scPmt4-MIR and O-mannosylated peptides, either in co-crystallization experiments or other biochemical assays. As is now, the statement on mannose binding/CBMs is rather speculative.

This is something we are actively working on both for the MIR domains and full-length protein, but due to the time-frame and complexity of the experiments will likely yield data for a future manuscript.

Line 190-196: Without experimental evidence for the “substrate/product peptide route”, this paragraph is rather speculative. I would suggest that the authors consider rephrasing “highly suggestive” with softer language.

We have removed “highly” from this statement.

Line 240: DDM extraction may, or may not, preserve the native lipid content of the assembly. This should be briefly stated for the readership.

This is now stated in the context of the observed cardiolipin binding (see below).

Line 244: Why was the identification of cardiolipin interesting? Was this expected or unexpected?

As a mitochondrial lipid, we do not expect Pmt4 to be associated with cardiolipin *in vivo* and think it is more likely to be present in our sample as an artifact of protein purification from total cellular membranes. Indeed, this has been observed previously for purified ER membrane proteins via the same lipidomics/native MS approach (McDowell et al., 2020). We have now clarified this point in the text.

Line 287-300: The result paragraph for the functional assay with the FUS-Axl2TMZZ substrate is not perfectly clear to me. What was the rationale? How many technical/biological replicates were done for Figure 5A? Why is there no loading control? These questions are rooted in my difficulties in interpreting the results in Figure 5A. If the assay based on assessing the absence/accumulation of the indicated band at ~68 kDa, why is there a difference in intensity between EV and WT and why is there no major accumulation in RRR and H219A/H637A (compared to EV)? The authors should consider clarifying.

As we found ctPmt4 expressed in *S. cerevisiae* has an altered substrate specificity relative to scPmt4, we tried several different substrate mannosylation assays and determined FUS-Axl2TM^{ZZ} to have the clearest readout. We performed three biological replicates and agree the chosen representative western blot was rather ambiguous; we have now replaced the western blot in Figure 5A with a replicate that shows these effects more clearly. In the absence of O-mannosylation (i.e. with the EV control), cells accumulate hypo-mannosylated FUS-Axl2TM^{ZZ} that runs on SDS-PAGE at ~70 kDa. We see that the wild type and H219A/H637A (which native MS showed is not impaired for Dol-P-Man binding) do not accumulate this species, whilst the RRR and G648L variants (which are impaired for Dol-P-Man binding) do accumulate hypo-mannosylated FUS-Axl2TM^{ZZ}, as per the EV control. The western blot of total membranes was repeated and analysed with respect to the amounts of ctPmt4-HA and Sec61 as loading control (Figure S12C).

Line 314-315: Does this argument also apply to other GT-C enzymes, e.g. DPY19Ls or TMTCs?

To answer this question, we superimposed our structure of ctPmt4 with structural models for other GT-C enzymes that use Dol-P-Man as a substrate. Strikingly, we found that human TMTCs and TMEM208 appear to also have a positively charged cavity in the position where we observe cytosolic Dol-P-Man binding, suggesting it is a general feature of O-mannosyltransferases. We have now added this observation to the text and Figure S12. According to these comparisons, DPY19, PIG and ALG enzymes do not appear to present a positively charged cavity in this exact position; indeed these (largely predicted) models generally differ in the length/arrangement at the cytosolic ends of the contributing TMDs.

Line 320: The discussion should perhaps briefly state that one current limitation is lack of insight into the Pmt4 dimer-substrate peptide complex, which precludes further insight into the catalytic cycle of the reaction.

We added this sentence to the discussion: "However, more detailed insights into the catalytic mechanism, substrate specificity and regulation of PMTs requires characterisation of the PMT4 homodimer bound to specific substrates, and more generally, the conformational landscape and putative higher order assemblies of PMTs."

Line 1045: Figure S1: Size exclusion elution profiles of ctPmt4 appear to migrate as monomers? It may be beneficial to indicate the migration pattern of molecular standards on the Superdex 200 Increase SEC. It may be further beneficial to indicate which elution fractions were used for CryoEM. It is unclear why ctPmt4 migrates at ~100 kDa (LMNG-CHS) and ~70 kDa (DDM) in the following SDS-PAGE analysis?

The manual for the Superdex 200 increase 10/30 GL column (Cytiva) shows that a 158 kDa standard elutes at 12.5 ml, similar to the volume for ctPmt4 (92 kDa). Nevertheless, we would argue that the soluble standard proteins used for calibration do not provide good markers for a membrane protein, whose hydrodynamic radius and elution volume are altered by the associated detergent micelle. In addition, proteins eluted between 10 and 15 ml are expected to cover the range ~44-440 kDa, indicating that species of quite different molecular weights migrate closely together. We think that the cryo-EM and native-MS together provide much better evidence for the oligomeric state of purified ctPmt4. At the reviewer's suggestion, we have now indicated the fraction(s) used for these experiments on the SEC profiles in Figure S1A.

We thank the reviewer for noticing the discrepancy in the gels in Figure S1A. Upon closer inspection, we realised the uncropped gel for ctPmt4 in LMNG-CHS was not perfectly straight in the image, which led to an offset of the markers. This has now been corrected showing that ctPmt4 runs consistently in both detergents just above the 70 kDa marker. The uncropped gels are provided in the Source Data file.